# GESR: A Geometric Evolution Model for Symbolic Regression

## Abstract

Symbolic regression is a challenging task in machine learning that aims to automatically discover highly interpretable mathematical equations from limited data. Keen efforts have been devoted to addressing this issue, yielding promising results. However, there are still bottlenecks that current methods struggle with, especially when dealing with complex problems containing various noises or with intricate underlying mathematical formulas. In this work, we propose a novel Geometric Evolution Symbolic Regression(GESR) algorithm. Leveraging geometric semantics, the process of symbolic regression in GESR is transformed into an approximation to an unimodal target in n-dimensional topological space. Then, three key modules are proposed to enhance the approximation: (1) a new semantic gradient concept, proposed to assist the exploration, which aims to improve the accuracy of approximation; (2) a new geometric search operator, tailored for approximating the target formula directly in topological space; (3) the Levenberg-Marquardt algorithm with L2 regularization, used for the adjustment of expression structures and the balance of global subtree weights to assist the proposed geometric semantic search operator. With the proposal of these modules, GESR achieves state-of-the-art accuracy performance on multiple authoritative benchmark datasets and demonstrates a certain level of robustness against noise interference. The implementation is available at https://anonymous.4open.science/r/12331211321-014D.

## 1 Introduction

Symbolic regression(SR) is a supervised machine learning method that learns interpretable mathematical expressions directly from a given dataset. SR provides us with the opportunity to automatically extract highly interpretable mathematical expressions that depict the underlying objective patterns in complex data from a mathematical perspective, rather than heavily relying on expert intuition and sensitivity to data. Compared to the numerical solutions generated through neural networks, quantitative mathematical expressions can provide more interpretability and generalization. Nowadays, SR has been widely applied in a variety of tasks, from scientific discovery (Makke & Chawla, 2024; Wang et al., 2023) to engineering applications (Angelis et al., 2023), including governing equations (Sun et al., 2022), finding fundamental physical laws (Udrescu & Tegmark, 2020), fault detection (Hale et al., 2022), and TCP congestion control (Sharan et al., 2022).

Genetic programming(GP) serves as the primary algorithm for SR. In GP, symbolic regression is addressed by evolving the expression trees within a population. With the iterative variation of the population and the guidance of fitness, the accuracy of expression trees gradually improves until a satisfactory level is achieved. GP has been proven to be very useful in many tasks, such as inferring physical laws. However, when the underlying formula is complex, its effectiveness still falls short of expectations due to the quite large search space.

With the exploration of symbolic regression and machine learning, numerous approaches are proposed to seek breakthroughs in terms of inference time, solution rate, accuracy, and model size. Deep Learning methods emerge due to their avoidance of the hyperparameter sensitivity problem in GP and their rapid inference time. SymbolicGPT (Valipour et al., 2021) uses deep language models like GPT to generate mathematical expressions for SR. AIFeynman (Udrescu et al., 2020) trains a neural network to estimate functional modularities. NGGP (Mundhenk et al., 2021) utilizes a hybrid approach to assist GP in generating a start population with neural networks. The end-to-end

transformer model for SR (Kamienny et al., 2022) also shows its capability on exploration and inference time. Reinforcement Learning is then employed to explore and discover expressions, whereas the Monte Carlo tree search method is utilized. DSR (Petersen et al., 2019) is a representative reinforcement learning method, which uses a learn policy to add notations to parse tree in turn while a neural network is utilized to generate symbolic distribution. Then, uDSR (Landajuela et al., 2022) is proposed as a comprehensive framework that hybridizes GP, AIFeynman, linear models, DSR, and large-scale pre-training into a single module and achieves state-of-the-art performance.

However, these methods still face challenges in effectively generating formulas from data characterized by intricate mathematical representations. Recently, the development of semantic genetic programming(SGP) (Pawlak et al., 2014; Chen et al., 2019; Huang et al., 2022) provides us with a fresh perspective. As we all know, the core idea of GP lies in the concept of iterative evolution, and the benefits of the "evolution" have been widely demonstrated. Nevertheless, the lack of explicitly guided cross-mutation processes severely hinders the improvement of the accuracy of formula evolution. Semantic genetic programming presents the unimodal landscape of symbolic regression, enabling the exploration of approximate unimodal solutions directly in the underlying topological space(named semantic space). Compared to the random exploration of traditional genetic programming, the approximation in the topological space exhibits significantly stronger guidance and superior search efficiency. However, suffering from dimension curse and sparse characteristics in semantic space, the search performance in the underlying semantic space rapidly slows down after several generations and the approximation process becomes convoluted. Severe tree bloating is also a bottleneck problem since SGP directly maps each step of semantic approximation to the modifications of expression trees without the tree structure information in the semantic space.

To address these issues, we propose a geometric evolution symbolic regression(GESR) algorithm. The proposed algorithm significantly improves the fitting performance of symbolic expression while effectively tackling concerns associated with tree bloating. Utilizing SRbench benchmark and SRSD benchmark for both black-box and scientific discovery symbolic regression, our method achieves state-of-the-art accuracy results across both benchmarks.

In summary, our work introduces several key contributions: (1) We introduce the concept of semantic gradients based on discovering the inconsistency of the approximation process in sub-semantic space and target semantic space, enabling accurate approximation of mappings between sub-objective spaces and objective spaces, thereby enhancing semantic fitting accuracy. (2) We develop a novel geometric semantic method capable of efficiently approximating target semantics in the sparse semantic space. (3) We also present a mechanism to assist the geometric semantic method in adjusting the generated expression structure and capturing the potential solution by zero-ing the weights of uninformative subtrees (via the LM algorithm). (4) Our proposed GESR consistently demonstrates promising performance across a diverse array of SRbench and SRSD benchmark datasets, as validated through rigorous ablation experiments. When compared to the baseline, GESR exhibits notably superior accuracy in scientific discovery tasks and showcases robust noise resistance capabilities.

## 2 RELATED WORK

**Genetic Programming**: genetic programming refers to an evolutionary algorithm instance applied to SR, in which a set of expression trees evolve iteratively under the guidance of fitness. Modules such as crossover, mutation, and selection are employed to evolve the population towards minimizing the distance from the optimal expression. Many popular frameworks use GP for symbolic regression (Cranmer, 2023; Burlacu et al., 2020; La Cava et al., 2018; de Franca & Aldeia, 2021b; Zhang et al., 2023) and a survey (Mei et al., 2022) is recommended for an overview of GP for SR. In this work, we inherited the concept of population evolution from GP, combining it with geometric semantic methods (Virgolin et al., 2019; Chen et al., 2019) and constant optimization.

**Geometric Semantic Evolution**: geometric semantic genetic programming(GSGP) (Vanneschi, 2016) is a type of algorithm that evolves the mathematical formula directly in n-dimensional topological space. With the semantic information, GSGP converts the traditional mutations into the operations with geometric properties in the underlying semantic space to approximate the unimodal target semantics and directly map this geometric operation to syntactic modifications of expression trees. Many works have been proposed to improve the performance (Pietropolli et al., 2022; Chen et al., 2019; Nguyen & Chu, 2020; Virgolin et al., 2019; Huang et al., 2024). However, due to the

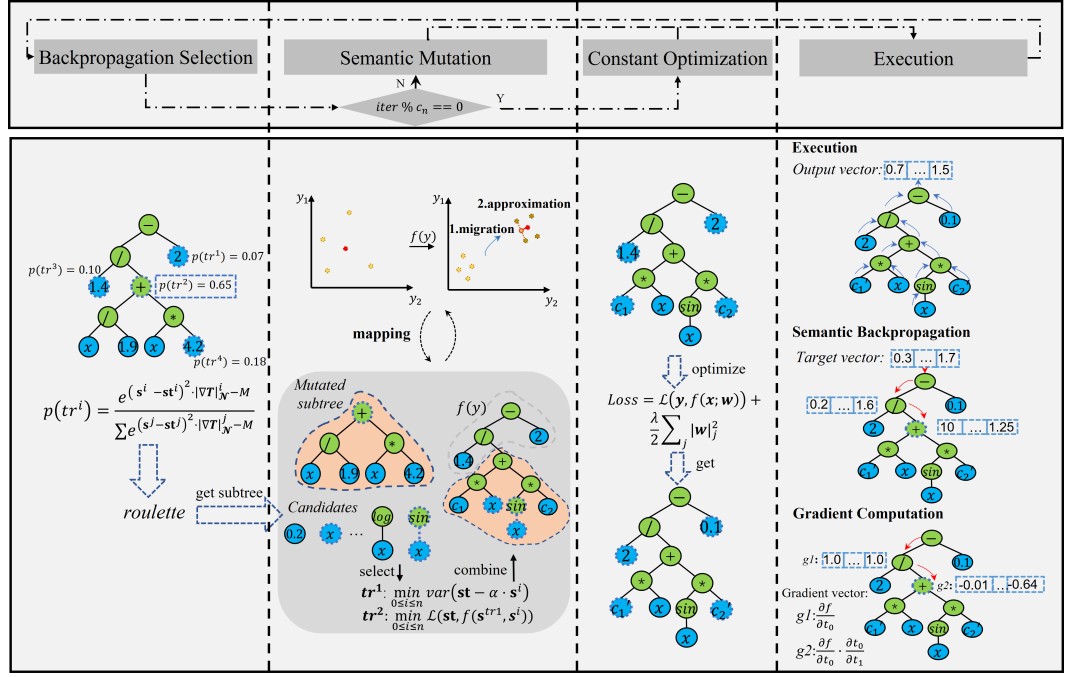

Figure 1: Schematic of the proposed method. *For each symbolic tree in the population, the GESR first selects the mutated subtree from a symbolic tree based on the calculated probabilities. The chosen subtree is then replaced by the generated combined tree through the semantics approximation in the topological space. Periodically, each tree in the population will be adjusted by the Weight Optimization module. The generated offspring is then executed to get the fitness value and the assisted information needed in the next iteration. After the execution, the evaluation and selection are performed and the selected symbolic trees form the new population for the next round iteration. These procedures are detailedly described in Section 3.*

high-dimensional nature of semantic space, these methods still struggle in the complex intermediate processes to reach the target semantics. Moreover, the direct mapping of geometric operations to syntax modifications also leads to severe tree bloating issues. Searching for target semantics in high-dimensional semantic space is still a challenging task.

## 3 METHODOLOGY

### 3.1 PRELIMINARY

Given a dataset $\mathcal{D} = \{(\mathbf{x}_i, \mathbf{y}_i)\}_{i \leq N}, (\mathbf{x}_i, \mathbf{y}_i) \in \mathbb{R}^d \times \mathbb{R}$, the main goal of symbolic regression (SR) methods is to find a symbolic expression $f^*$ that minimizes the theoretical risk of the input-output mapping of the dataset $\mathcal{D}$:

$$f^* = \arg\max_{f \in \mathcal{F}} \mathbb{E}_{\mathcal{D} \sim \Omega(\mathcal{D})}[\mathcal{L}(f, \mathcal{D})] \tag{1}$$

Semantic genetic programming (SGP) vectorizes the output of the symbolic expression related to the dataset $\mathcal{D}$ as a semantic vector, with which the iterative improvement of the symbolic expression can be achieved through semantic target approximation in topological space. The semantic vectors of possible solutions constitute semantic space, where each symbolic expression is mapped to a semantic vector point. In this paper, we propose a geometric semantic genetic programming algorithm, which utilizes geometric properties describing spatial relationships between symbolic expressions in the n-dimensional semantic space, where n is equivalent to the size of the training set. By directly approximating the target semantic with a geometric method in the semantic space and mapping the corresponding geometric operations to the syntactic modifications of symbolic expression, newly

obtained symbolic expression can be closer to the target. For example, the midpoint of the two semantic points $\mathbf{s}^1$ and $\mathbf{s}^2$ that represent two symbolic expression trees $tr^1$ and $tr^2$ respectively in the semantic space can be mapped to a new expression tree: $\mathbf{s}' : 1/2\mathbf{s}^1 + 1/2\mathbf{s}^2 \rightarrow tr' : 1/2 * tr^1 + 1/2 * tr^2$.

In our method, geometric semantics is used to guide the mutation, as shown in Figure 1. However, although geometric semantic mutation improves search efficiency and accuracy, frequent linear combinations of expressions often lead to severe tree bloating. Therefore, we backpropagate the target semantics to sub-expression, combined with semantic gradients and strict limitations, to guide the mutation process in the sub-semantic space. The linear combination of sub-expressions also leads to a significant increase in the proportion of constants. While geometric semantic mutation based on backpropagation is restricted to only adjusting local constants, it is difficult to balance the optimization of global constants. Thus, we incorporate the Levenberg-Marquardt optimization algorithm with L2 regularization to periodically adjust global constants, which also enables the smoothing of formula curves. By the way, since we assign a constant (weight) to each combination subtree, there is a natural characteristic for our method to use a continuous optimization algorithm to further adjust the overall tree structures.

## 3.2 SEMANTIC GRADIENT

Semantic backpropagation is achieved by inversely calculating the expression tree (Krawiec & Pawlak, 2013). As the target semantics are reversed into sub-target semantics, the semantic space is also mapped to a sub-semantic space. Thus the solution in the semantic space can be found by searching within the sub-semantic space. In this section, a semantic gradient concept is further proposed to assist the exploration in semantic space, which comes from the following observation: **The approximation process of the sub-semantic space cannot be equivalently propagated to the root semantic space.** Figure 2 is an example for better understanding. While the target semantics can be backpropagated to the sub-semantic space, the approximation process will be seriously affected by the calculation path of the mutation node, where the calculation path serves as the mapping function $f(y)$. With $f(y)$, the evaluation based on the Euclidean distance between the semantics $\mathbf{s}$ of subtree $tr$ and the sub-target semantics $\mathbf{st}$ in the sub-semantic space: $\sqrt{\sum_{i=0}^{n} (\mathbf{s}_i - \mathbf{st}_i)^2}$, is transformed into $\sqrt{\sum_{i=0}^{n} (f(\mathbf{s}_i) - f(\mathbf{st}_i))^2} \Rightarrow \sqrt{\sum_{i=0}^{n} (w \cdot 2\mathbf{x}_i)^2 \cdot (\mathbf{s}_i - \mathbf{st}_i)^2}$. The uncertain vector $\mathbf{x}$ makes each dimension of the approximation result from the sub-semantic space deviate in the target semantic space, which significantly reduces the accuracy of semantic approximation based on backpropagation. To address this issue, we introduce the concept of semantic gradients to approximate the mapping

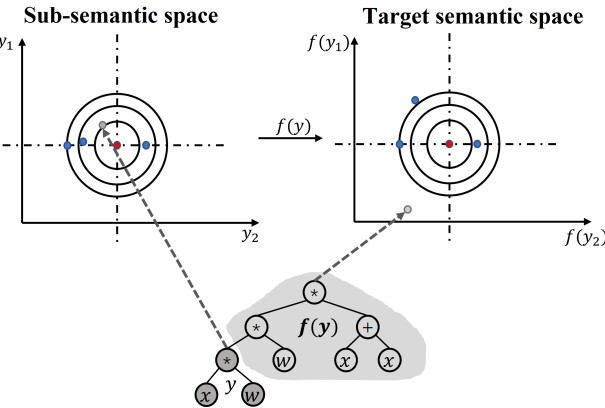

Figure 2: Spatial transformation under nonlinear mapping function f(y). With f(y), the Euclidean distance between semantic points in the sub-semantic space changes when mapped to the target semantic space.

relationship between the sub-semantic space and the target semantic space. As illustrated in Figure 3, starting from the root node of the expression tree, the semantic gradients of the mutated node are computed along the semantic backpropagation path, thereby obtaining the gradient vector $\nabla T$ for the

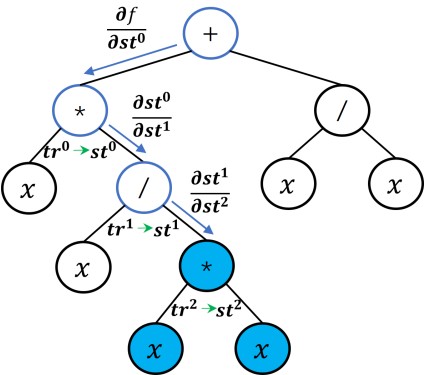

Figure 3: The process of semantic gradient computation. The semantic vectors $\mathbf{s}^i$ on the backpropagation path are replaced by the corresponding sub-target semantic $\mathbf{st}_i$.

$j^{th}$ node. The representation of $\nabla T$ is as follows:

$$\nabla T =< \frac{\partial f_0}{\partial \mathbf{s}_0^j}, \ldots, \frac{\partial f_n}{\partial \mathbf{s}_n^j} >=< \frac{\partial f_0}{\partial \mathbf{s}_0^0} \cdot \frac{\partial \mathbf{s}_0^0}{\partial \mathbf{s}_0^1} \cdots \frac{\partial \mathbf{s}_0^{j-1}}{\partial \mathbf{s}_0^j}, \ldots, \frac{\partial f_n}{\partial \mathbf{s}_n^0} \cdot \frac{\partial \mathbf{s}_n^0}{\partial \mathbf{s}_n^1} \cdots \frac{\partial \mathbf{s}_n^{j-1}}{\partial \mathbf{s}_n^j} > \quad (2)$$

Where $\mathbf{s}_i^j$ denotes the $i^{th}$ dimension of the semantics of the $j^{th}$ subtree. The partial derivative values across various semantic dimensions indicate the changing rate of the nonlinear mapping function $f(tr^j)$ with respect to the semantics of the subtree $tr^j$. Specifically, the absolute partial derivative values across each semantic dimension serve as an indicator of the degree of influence that deviations of $tr^j$ within the corresponding sub-semantic space have on the deviations of $f(tr^j)$ within the corresponding dimension of the target semantic space. Thus, the normalized value $|\nabla T|_{\mathcal{N}}^j$ serves as the weight of each dimension in the sub-semantic space. As the gradient vector is affected by the semantics of the mutated node which is unfixed, we use the sub-target semantics $\mathbf{st}$ instead of the current mutated node semantics $\mathbf{s}$ when computing the semantic gradient:

$$\nabla T =< \frac{\partial f_0}{\partial \mathbf{st}_0^j}, \frac{\partial f_1}{\partial \mathbf{st}_1^j}, \ldots, \frac{\partial f_n}{\partial \mathbf{st}_n^j} > \quad (3)$$

Where $\mathbf{st}_i^j$ refers to the sub-target semantics of the $i^{th}$ dimension of the $j^{th}$ subtree.

### 3.3 GEOMETRIC SEMANTIC METHOD

Due to the high-dimensional nature of the semantic space, in which the dimensions equal to the training set size, the sparsity of semantic points resulting from high-dimensional features makes it difficult for geometric semantic mutation strategies to search for target semantics in the semantic space. To quickly approximate the target semantics in the semantic space, two steps are conducted as follows. First, as the distribution of candidates in the semantic space is sparse, we scale each candidate semantics $\mathbf{s}$ to the perpendicular projection of the sub-target semantics onto the line connecting the semantics point and the origin point. This step aims to migrate the distribution of candidate semantics $\mathbf{s}$ to the sub-target semantics $\mathbf{st}$ around the semantic space, as shown below:

$$\mathbf{s}' = \alpha \mathbf{s} \quad (4)$$

where, $\alpha$:

$$\alpha = \frac{\mathbf{s} \cdot \mathbf{st}}{\mathbf{s} \cdot \mathbf{s}} = \frac{\sum_i^n \mathbf{s}_i \cdot \mathbf{st}_i}{\sum_i^n \mathbf{s}_i \cdot \mathbf{s}_i} \quad (5)$$

Subsequently, we adopt $var(\mathbf{st} - \alpha \cdot \mathbf{s}) = \sum_{i=1}^n ((\mathbf{st}_i - \alpha \cdot \mathbf{s}_i) - E(\mathbf{st}_i - \alpha \cdot \mathbf{s}_i))^2$ to rank the candidate set, instead of utilizing the Euclidean distance metric, aiming to get the subexpressions that are more similar in functional form rather than a smaller scalar distance. The possible constant deviation can be eliminated with the constant node in the candidate set, as detailed in Appendix B. The approximated candidate set then is sorted based on the Euclidean distance $dis(\mathbf{s}_i', \mathbf{st})$ between the semantics of candidates and the sub-target semantics. After that, we select multiple pair of

candidates in order and use the least squares method to combine them pairwise to find the linear combination of candidates that best approximates the target semantics. For the selection of $tr^1$, we select the $top\text{-}t$ subtrees within the sorted set as candidates. For the selection of $tr^2$, the following $m$ subtrees for each $tr^1$ are chosen to form the candidate set. Then, the obtained candidate subtrees from $tr^1$ set and $tr^2$ set are linearly combined to minimize the distance $\mathcal{L}(\mathbf{st}, f(\mathbf{s}^{tr1}, \mathbf{s}^{tr2}))$, which can be represented as below with Eq 4:

$$\mathbf{s} = \alpha_1 \cdot (1 - k) \cdot \mathbf{s}^{tr1} + \alpha_2 \cdot k \cdot \mathbf{s}^{tr2} \tag{6}$$

Here, by utilizing the least squares method to minimize the loss function $\mathcal{L}(\mathbf{st}, f(\mathbf{s}^{tr1}, \mathbf{s}^{tr2})) = \sum(\mathbf{st}_i - f(\mathbf{s}_i^{tr1}, \mathbf{s}_i^{tr2}))^2 \cdot |\nabla T|_{\mathcal{N},i} = \sum(\mathbf{st}_i - \alpha_1 \cdot (1 - k) \cdot \mathbf{s}_i^{tr1} - k \cdot \alpha_2 \cdot \mathbf{s}_i^{tr1})^2 \cdot |\nabla T|_{\mathcal{N},i}$, we can obtain $k$ as follows:

$$k = \frac{|\nabla T|_{\mathcal{N}} \odot (\alpha_2 \mathbf{s}^{tr2} - \alpha_1 \mathbf{s}^{tr1}) \cdot (\mathbf{st} - \alpha_1 \mathbf{s}^{tr1})}{|\nabla T|_{\mathcal{N}} \odot (\alpha_2 \mathbf{s}^{tr2} - \alpha_1 \mathbf{s}^{tr1}) \cdot (\alpha_2 \mathbf{s}^{tr2} - \alpha_1 \mathbf{s}^{tr1})} \tag{7}$$

In the Eq 7, $|\nabla T|_{\mathcal{N}}$ represents the normalized vector of the absolute semantic gradient. With Eq 4, 5, 6, 7, a new combined tree can be generated: $tr' = \alpha_1 \cdot (1 - k) \cdot tr^1 + \alpha_2 \cdot k \cdot tr^2$. For the generated candidates $\{tr'^1, tr'^2, ..., tr'^n\}$ after pairwise linear combinations, we first filter the candidate set based on the Euclidean distance $\mathcal{L}(\mathbf{st}, \mathbf{s}')$ between the semantics $\mathbf{s}'$ of each combined candidate subtree $tr'$ and the sub-target semantics $\mathbf{st}$ in the root semantic space. Considering that the Euclidean distance used as the evaluation criterion is a scalar value, which loses the vector information in the semantic space. Simply pursuing a decrease in distance may lead to a loss of diversity and trap into local optima. Therefore, an additional parameter $\lambda$ is used to fully explore the solution space:

$$\mathcal{L}(\mathbf{st}, \mathbf{s}') \leq \mathcal{L}(\mathbf{st}, \mathbf{s}^c) \cdot (1 + \lambda)$$
$$\Rightarrow \sum \left( (\mathbf{st}_i - \mathbf{s}_i')^2 \cdot |\nabla T|_{\mathcal{N},i} \right) \cdot (1 + \lambda) \leq \sum \left( (\mathbf{st}_i - \mathbf{s}_i^c)^2 \cdot |\nabla T|_{\mathcal{N},i} \right) \tag{8}$$

Where $\mathbf{s}^c$ refers to the semantics of the original subtree, and the candidates that do not satisfy the Eq 8 are filtered out. Moreover, we incorporate a variance-based constraint method to ensure smoother function fitting, thereby avoiding potential issues of local overfitting caused by uneven data distribution or significant differences of output values within the dataset. The constraint is formulated as follows:

$$\frac{var(\mathbf{st} - \mathbf{s}^c)}{var(\mathbf{st} - \mathbf{s}')} = \sum_{i=1}^{n} \frac{\left( (\mathbf{st}_i - \mathbf{s}_i^c) - E(\mathbf{st} - \mathbf{s}^c) \right)^2}{\left( (\mathbf{st}_i - \mathbf{s}_i') - E(\mathbf{st} - \mathbf{s}') \right)^2} \geq 1 + \lambda \tag{9}$$

The remaining candidate set that satisfies both Eq 8 and Eq 9 is evaluated in the final step. Since the semantics is solely constituted by output vectors, lacking tree structure information, which results in a severe expansion problem when translating the semantic space approximation to expression trees. Inspired by the SPL approach (Sun et al., 2022), the final evaluation function is calculated using the following formula:

$$\mathcal{R} = \frac{\eta^l}{1 + \sqrt{\sum_{i=1}^{n}(\mathbf{st}_i - \mathbf{s}_i')^2 \cdot |\nabla T|_{\mathcal{N},i}}} \tag{10}$$

Where $\eta$ is the discount factor promoting concise trees, $l$ represents the mutated subtree size. We only replace the mutated subtrees where $\mathcal{R}$ is smaller than the original $\mathcal{R}^c$. It is worth noting that the mutated subtree size is calculated by the inner node size instead of the tree size to ensure the fairness of each function symbol. By employing this evaluation function, our method encourages finding more concise expressions while minimizing the Euclidean distance.

### 3.4 SEMANTIC POINT TOURNAMENT SELECTION

In selecting a subtree in a symbolic tree to mutate, the traditional random selection strategy without any prior information reduces search efficiency. We integrate semantic information with the semantic gradient into the mutated subtree selection strategy to guide the selection of mutated subtrees, where the Euclidean distance is chosen as the basic metric for assessing the optimization potential of candidate mutated subtrees. By computing the Euclidean distance between the semantics of the candidate mutated subtree and the sub-target semantics, and mapping it through the semantic gradient to the root semantic space, we obtain a scalar expression of the potential for each candidate mutated subtree. Then, we utilize softmax to convert the mapped Euclidean distance of each candidate mutated

subtree into probabilities and use a tournament selection strategy to choose the final mutated subtree from the candidate subtrees. The selected probability $p(tr^i)$ of each candidate $tr^i$ is computed by:

$$p\left(tr^i\right) = softmax\left(\mathbf{r}^i\right) = \frac{e^{(\mathbf{st}^i - \mathbf{s}^i)^2 \cdot |\nabla T|_{\mathcal{N}}^i - M}}{\sum e^{(\mathbf{st}^j - \mathbf{s}^j)^2 \cdot |\nabla T|_{\mathcal{N}}^j - M}} \tag{11}$$

Where $\mathbf{r}^i = (\mathbf{st}^i - \mathbf{s}^i)^2 \cdot |\nabla T|_{\mathcal{N}}^i$ and $M = \max_{0 \leq i \leq n} \mathbf{r}^i = \max_{0 \leq i \leq n}((\mathbf{st}^i - \mathbf{s}^i)^2 \cdot |\nabla T|_{\mathcal{N}}^i)$ is used to prevent data overflow. It is worth noting that we only randomly select a few subtrees as candidates in the expression tree in each generation to make full exploration while ensuring randomness, since the mismatched parts of an expression may be scattered across multiple branches of the expression tree.

## 3.5 Weight Optimization

The optimization of geometric semantic mutation lies solely in the adjustment of local tree structures, and it is worth noting that our geometric semantics method provides weight coefficients for each subtree(Equation 6). Differs from the traditional methods that use a continuous optimization algorithm to optimize the constants (Kommenda et al., 2020; De Melo et al., 2015), the presence of these weight coefficients allows us not only to adjust each subtree module at any time but also, more importantly, to automatically adjust the overall structure by optimizing constants and removing unsuitable subtrees (by setting the corresponding subtree's weight coefficient to 0.0, when the weight coefficient is approximated to nearly zero(<1e-6 in our method)). Therefore, we employ the Levenberg-Marquardt(LM) (Moré, 2006) algorithm to perform constant optimization on the expressions at regular intervals. The LM algorithm utilizes Euclidean distance as the loss function. Additionally, considering that expressions generated based on geometric semantics often contain numerous constants, we have added L2 regularization to smooth the function curve, as shown below:

$$Loss = \mathcal{L}\left(\mathbf{y},\ f\left(\mathbf{x}; \mathbf{w}\right)\right) + \beta/2 \sum_j |\mathbf{w}_j|^2 \tag{12}$$

Following with LM algorithm, we can optimize the constants $\mathbf{w}$ through the formula:

$$\mathbf{w}^{n+1} = \mathbf{w}^n - \left(\mathbf{J}_r^T \mathbf{J}_r + \mu \mathbf{I}\right)^{-1} \left(\mathbf{J}_r^T r + \beta|\mathbf{w}|\right) \tag{13}$$

Where $\mathbf{J}_r$ is the Jacobian matrix of constants, $\mathbf{I}$ is the identity matrix, $\mu$ is the penalty factor, $\beta$ is the hyperparameter, and $\mathbf{w}$ represents constants within the expression.

# 4 Experiments

## 4.1 Basic Benchmarks

We assess GESR on two mainstream large-scale datasets: the SRBench benchmark (La Cava et al., 2021) which includes both real-world and ground-truth problems, and the SRSD benchmark (Matsubara et al., 2022) for scientific discovery. A total of 25 baseline methods are used for performance comparison in the experiment. It is worth noting that these baseline methods use multiple parameter sets and find the most suitable parameter settings through a half-grid search. However, considering the significant impact of parameters on algorithm performance, to demonstrate the superiority of our method and reduce the influence of parameter selection, the hyper-parameters of our method remain the same on all datasets to control the influencing factors. The detailed hyper-parameter setting and the brief introductions to the baseline methods are provided in Appendix A and C respectively. The PMLB dataset includes 46 real-world black-box datasets, including physics, home price, etc., 76 synthetic datasets including Friedman datasets, 130 Strogatz datasets, and Feynman datasets. Among them, real-world datasets and Friedman datasets(form the black-box datasets) are used for overall evaluations, Friedman datasets are used for ablation analysis, Strogatz and Feynman datasets with different levels of Gaussian noise are used for robustness verification. In addition, the SRSD benchmark with 120 scientific discovery datasets that contain dummy variables is also used to test the effectiveness of GESR, in which the properties of the formula and the variables are carefully reviewed to ensure a realistic sampling value range. Our method is assessed with a popular metrics: $R^2$-score (La Cava et al., 2021).

Table 1: SRSD+ Dummy Variables: Accuracy solution rate($R^2 > 0.999$) on scientific discovery datasets with different complexity.

| Group | gplearn | AFP | AIF | DSR | E2E | uDSR | PySR | GESR |
|---|---|---|---|---|---|---|---|---|
| **Easy** | 0.00% | 20.0% | 6.67% | 76.7% | 16.7% | 53.3% | 20.0% | 100.0% |
| **Medium** | 0.00% | 5.00% | 0.00% | 45.0% | 12.5% | 37.5% | 10.0% | 87.5% |
| **Hard** | 0.00% | 4.00% | 0.00% | 22.0% | 10.0% | 12.0% | 2.0% | 58.0% |

## 4.2 EFFECTIVENESS OF GESR

We evaluated our method on the SRSD benchmark that contains 120 scientific discovery datasets with dummy variables, to assess the performance in tackling scientific discovery problems. As shown in Table 1, our method achieved accuracy solution rates($R^2 > 0.999$) of 100.0%, 87.5%, and 58% on the easy, medium, and hard datasets, which means that our method gets the improvements of 23.3%, 42.5%, 36% over the second-ranked baseline method respectively. The experimental result shows the powerful search ability of our geometric semantic method in approximating target semantics, especially in addressing complex scientific discovery problems.The further experimental results on the SRSD benchmark are presented in Appendix D.3, and the qualitative analysis of the GESR has been discussed in Appendix D.4.

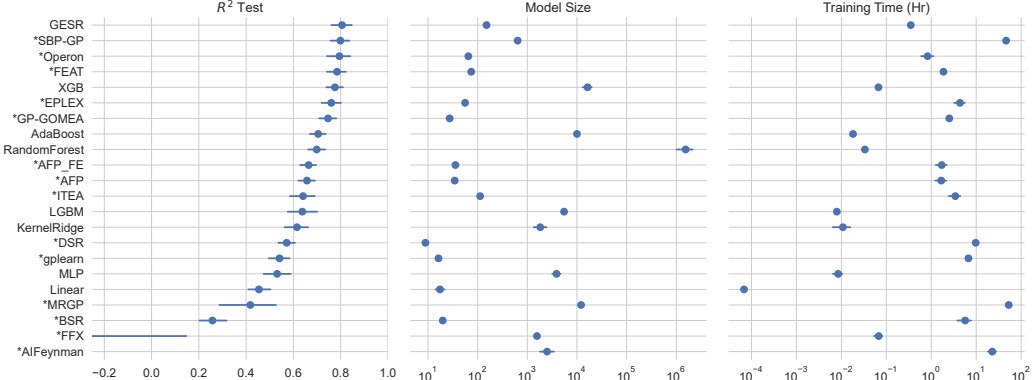

Figure 4: SRBench-generated comparisons of $R^2$ test(left) and model size(right) on 120 black-box datasets.

Figure 4 presents the performance of GESR on the SRBench benchmark. In terms of overall accuracy, our method outperforms all other algorithms. Additionally, our method also exhibits advantages in model size and shows substantial improvements compared to previous semantics-based approaches (SBP-GP).

In addition to conducting large-scale evaluations on the scientific discovery problems within the SRSD benchmark and real-world datasets from the SRBench benchmark, we have also tested our method on several common datasets (Nguyen, Livermore). Further experimental details have been provided in Appendix C.

## 4.3 FURTHER ANALYSIS

In our method, symbolic expression is generated through the direct data fitting at the semantic level. This may give the impression that the GESR is weak in noise robustness ability. To further explore the adaptability of the GESR to noise and the capability to solve the scientific discovery datasets, we conducted additional experiments on the synthetic dataset Strogatz with different levels of Gaussian noise.

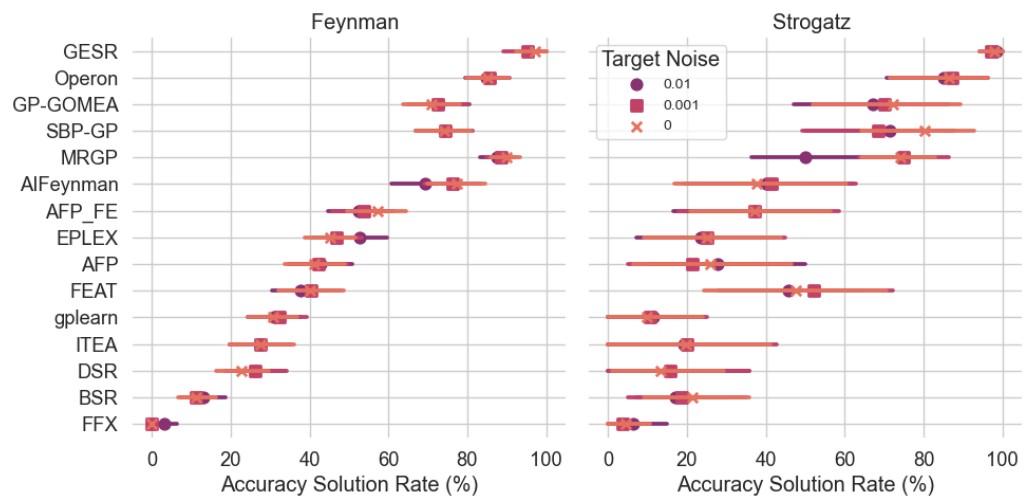

Figure 5: Accuracy solution rate($R^2 > 0.999$) on Feynman datasets(left) and Strogatz datasets(right) of SRBench benchmark with different noise levels.

As depicted in Figure 5, our method gets state-of-the-art performance. Compared to the black-box datasets with the unknown underlying data generated function containing substantial irregular noise and dummy variables, there a more pronounced distinctions among the baseline methods for the capability to solve the scientific discovery datasets. It can be observed that the Gaussian noise has a negative impact on the accuracy solution rate of most methods. The stable accuracy solution rate of GESR than other methods for different levels of Gaussian noise demonstrates the certain degree of the noise-resistant capabilities of our method. The specific performance of our method on each dataset is provided in Appendix D.6, in which it can be observed that our method outperforms others on most datasets in terms of median $R^2$, mean $R^2$, and solution rate.

## 4.4 ABLATION ANALYSIS

In GESR, several components have been integrated. To validate the necessity of each component, we conduct ablation experiments on the Friedman dataset to individually examine the effects of the semantic gradient strategy, semantic mutation point selection strategy, geometric semantic approximation strategy, and constant optimization. Additionally, the full GESR method is used as a baseline. Table 2 shows the average $R^2$ score, average $R^2$ rank, and average model size on Friedman

Table 2: The experimental results on Friedman datasets with the removal or replacement of each component. GESR-opt, GESR-slt, and GESR-gradient correspond to the removal of the constant optimization module, the semantic mutation point selection module, and the semantic gradient module. GESR-mutation refers to the replacement of the geometric semantic approximation module to linear scale strategy in (Virgolin et al., 2019). The GESR-baseline refers to the complete GESR method.

|  | GESR-opt | GESR-slt | GESR-gradient | GESR-mutation | GESR-baseline |
|---|---|---|---|---|---|
| Avg rk | 2.62 | 1.39 | 2.53 | 2.32 | 1.11 |
| Avg $R^2$ | 0.929 | 0.939 | 0.901 | 0.929 | 0.947 |
| Avg size | 185 | 161 | 123 | 120 | 134 |

datasets with the lack or replacement of each component. Generally from the experimental results, the removal or replacement of any component shows a negative impact on our method, no matter the average $R^2$ or the average rank, which highlights the necessity of the proposed components. To further test whether there are statistically significant difference in performance with the removal of replacement of each component, a Wilcoxon signed-rank test (Wilcoxon, 1992) is performed on

the median experiment results. Compared with the GESR-baseline, GESR-opt, GESR-slt, GESR-gradient, and GESR-mutation yields p values below 1e-6, 1e-2, 1e-7, and 1e-6, respectively, indicating significant differences ($p < 0.01$) with and without the inclusion of the proposed weight optimization module, semantic point tournament selection module, geometric semantic method, and semantic gradient module respectively.

Due to the nonlinear mapping of semantic space, the importance of each dimension in the sub-semantic space is not equivalent. The absence of semantic gradients impairs the accuracy and effectiveness of the semantic variation module. Therefore, compared to the GESR-baseline, the GESR-gradient without the semantic gradient module exhibits a significant drop in the average $R^2$ metric. Another significant drop in accuracy performance is observed for GESR-mutation, which demonstrates the superiority of the geometric semantic approximation module. In terms of average ranking, GSR-Baseline also shows the better stability over performance across different datasets. The details and roles of each module have been further discussed in Appendix B.

## 5 CONCLUSION

In this work, a novel geometric evolution model is proposed for SR to try to improve the spatial approximation accuracy while ensuring the interpretability. In GESR, a geometric semantic mutation method is introduced as a geometric approximation strategy to search for expressions with optimal accuracy in the semantic space, using a semantic gradient vector as an auxiliary tool for addressing semantic space mapping issues. A semantic-based mutation point selection method is further introduced to efficiently identify sub-expressions that need improvement. Taking into account the inherent advantage of linear combinations within the semantic space, we further incorporate the Levenberg-Marquardt algorithm as a key module for globally adjusting and optimizing expressions constants. The state-of-the-art accuracy performance achieved on both the SRSD and SRBench benchmark datasets demonstrates the effectiveness of our approach in real-world and scientific discovery tasks.

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

APPENDIX

## A    BRIEF DESCRIPTIONS OF THE BASELINE METHODS

In this section, for the convenience of reference and comparison, we provide the short descriptions of the baseline methods, including the 14 original SRBench baseline methods and 5 SRSD baseline methods that used in Section 4. The original papers are refered for the additional details.

- **Unified Framework for Deep Symbolic Regression(uDSR)**: uDSR Landajuela et al. (2022) is a modular framework which integrate multiple different SR solution strategies in an attempt to complement each other so as to maximize the advantages of each strategy.

- **Deep symbolic regression(DSR)**: DSR Petersen et al. (2019) leverages the deep reinforcement learning to search solution for symbolic regression, which employs a neural network to represent token distribution and a reinforcement learning strategy to train the netowrk.

- **AIFeynman(AIF)**: AIFeynman Udrescu et al. (2020) is a symbolic regression method that leverages neural network, graph modularity, hypothesis testing and normalizing flows to improve the accuracy towards harder problems and the robust towards noise.

- **SR with Non-linear least squares(Operon)**: OperonBurlacu et al. (2020) is a GP framework that aims to achieve efficient implement and provide a completely out-of-the-box solution for symbolic regression.

- **Semantic backpropagation genetic programming(SBP-GP)**: SBP-GPVirgolin et al. (2019) is a semantic genetic programming method. SBP-GP utilizes linear-scaling technology to shrink the candidate subtree to approximate the output of variation subtree to the sub-target semantics.

- **End-to-end Symbolic Regression with Transformers(E2E)**: E2E Kamienny et al. (2022) is an end-to-end symbolic regression method based on transformer to generate the full mathematical expression, which is in advantage of inference time.

- **PySR**: PySR Cranmer (2023) is an open-source library, in which the multi-population evolution strategy with the evolve-simplify-optimize loop is the kernel search algorithm.

- **Multiple regression genetic programming(MRGP)**: MRGP Arnaldo et al. (2014) is a genetic programming method which decouples and linearly combines the subexpressions of a program by multiple regression on the target variables.

- **$\epsilon$-lexicase selection(EPLEX)**: EPLEX La Cava et al. (2019) is a genetic programming method with $\epsilon$-lexicase selection strategy that improves the parent selection process, in which the training cases are used as evaluation value individually instead of aggregate all of the cases.

- **Feature engineering automation tool(FEAT)**: FEAT Cava et al. (2019) is a method that for learning and optimizing the interpretable representations, in which neural networks used as features are represented as syntax trees and an archive of representations that characterize the accuracy-complexity trade-offs are maintained to assist in the generalization and interpretation.

- **Fast function extraction(FFX)**: FFX McConaghy (2011) is a non-evolutionary technique leveraging pathwise regularized learning to generate a set of solutions that trade off error versus complexity, which improves the speed, scalability, and deterministic behavior.

- **GP version of the gene-pool optimal mixing evolutionary algorithm(GP-GOMEA)**: GP-GOMEA Virgolin et al. (2021) is a genetic programming method with improved linkage learning approach imposing a strict limitation size, which focuses on accuracy and small solutions.

- **Age-fitness Pareto optimization(AFP)**: AFP Schmidt & Lipson (2010) is a multi-objective genetic programming method, which evolves population on age-fitness pareto front to avoid premature convergence.

- **AFP with co-evolved fitness estimates(AFP-FE)**: AFP-FE Schmidt & Lipson (2009) is an improved AFP method with a new co-evolved fitness estimation method.

- **Bayesian symbolic regression(BSR)**: BSR Jin et al. (2019) is a Bayesian framework that employs Markov chain Monte Carlo method to incorporate prior knowledge and sample symbolic trees from posterior distribution, which shows advantages in interpretability, utilization to prior knowledge, and effective memory usage.
- **ITEA**: ITEA de Franca & Aldeia (2021a) is a mutation only evolutionary algorithm with a new individual representation way, in which each mutation process is achieved by randomly selecting among six heuristic mutation operators.

## B  GENERALIZE FRAMEWORK OF GESR

In Algorithm 1, we provide the detailed pseudocode for describing the framework of GESR. The population and semantic library are initialized in Lines 1-2, and the detailed parameter settings are provided in Appendix C.1. In GESR, Two mutation strategies are employed: geometric semantic approximation as the primary variation method(Line 9), and a random replacement strategy for population exploration(Line 10), both performed probabilistically in each generation. The Levenberg-Marquardt algorithm is periodically executed to adjust the global weight parameters of the expression(Line 5). Subsequently, mutated expressions are executed, and selected of nodes for the next mutation(Line 14). Additionally, at regular intervals, several subtrees from top-t expression trees in the population are randomly selected to supplement the semantic library for transfer learning the useful features (Lines 16-19).

---

**Algorithm 1:** Geometric evolution symbolic regression.

**Input** : Symbolic regression problem $\mathcal{P}$, consists of tabular data$(X, y)$
**Output** : Best fitting expression
1 Initialize population $\mathcal{P}$;
2 Initialize semantic library $\mathcal{D}_s$;
3 **for** *each iteration $i$* **do**
4    **if** *$i$ is equal to the interval of constant optimization* **then**
5       $\mathcal{C} \leftarrow$ optimize population $\mathcal{P}$ with Levenberg-Marquardt algorithm
6    **else**
7       $s \sim U(0, 1)$;
8       **if** *$s < 0.95$* **then**
9          $\mathcal{C} \leftarrow$ approximate the target semantics of $\mathcal{P}$ with geometric semantic mutation
            method.
10       **else**
11          $\mathcal{C} \leftarrow$ population variation with random mutation method.
12       **end**
13    **end**
14    Execute the symbolic tree in $\mathcal{C}$;
15    Select $\mathcal{P}$ with tournament strategy to form the new population$\mathcal{P}'$;
16    **if** *$i$ is equal to the interval of semantic library update* **then**
17       $\mathcal{T} \leftarrow$ select tok-n symbolic tree from $\mathcal{P}$;
18       update $\mathcal{D}_s$ with sub-trees from $\mathcal{T}$;
19    **end**
20 **end**

---

In the geometric semantic mutation strategy, for each expression tree in the population, we first select a subtree to be mutated from several candidate nodes based on Eq 11. Subsequently, a specified number of candidates are randomly selected from the semantic library, and linear combination and filter through Eq 4 - 10 are performed to replace the mutated subtree. It is worth noting that the representation of the subtree obtained by our geometric semantic method typically takes on the form $k_1 p_1 + k_2 p_2$, When $k_1$ or $k_2$ is 0.0, it yields $k \cdot p$ or 0. However, this resulting subtree lacks a combined representation of constants and features, i.e., $k_1 p_1 + c$. Therefore, for each expression tree in the population, we insert an additional constant node "1" into the candidate so that this combinatorial subtree can generate such a representation.

In the random replacement strategy, two traditional GP methods are employed: random subtree replacement and random node replacement. With a tournament selection strategy in small size, this

strategy can provide diversity in the evolution. Furthermore, to reduce the tree size, the limited subtree depth generated by the random subtree replacement strategy is less than or equal to the original subtree.

## C  EXPERIMENTAL SETUP

### C.1  HYPERPARAMETER SELECTION

Table 3: Hyperparameter setting

| hyper-parameter | Value |
|---|---|
| Population size | 500 |
| Generations | 200 |
| Initial tree depth | 1-3 |
| Depth limitation | 8 |
| Weight optimization interval | 20 |
| Semantic library size | 5000 |
| Geometric semantic mutation rate | 0.5 |
| Tournament size | 2 |
| Library update interval | 10 |
| Semantic candidate size | 200 |
| Top-t for the candidate selection | 1 |
| Top-m for the candidate selection | 200 |
| Functions | $< +, -, *, \%protected, \sin, \cos, \log, \exp >$ |

The hyperparameters for GESR are listed in Table 4. In this work, our method aims to directly approximate the output of the target expression numerically and then map the operation to the formula in the population. To mitigate the impact of poor initial expression structure, we employ a parameter setting with a large population and small depth, allowing for broader coverage of initial expression structures and increased fault tolerance. A smaller tournament size also facilitates group exploration. For variation rate, we primarily utilize geometric semantic search methods while maintaining a certain probability of random variation. The significance of random variation lies in its facilitation of early-stage population exploration, prevention of local optimization due to the solidification of population expressions in later stages, and selection of new expression structures into the semantic library. The updated pattern for the semantic database involves high-frequency sampling with small amounts to sample elite individuals' expression structures at each stage of evolution. Each semantic database samples 20 subtrees from the top 10 individuals' semantics and randomly replaces these subtrees within the database. During evolution, our function set consists of 8 tuples <+, -, *, %protected, sin, cos, log, exp> protected by interval arithmeticKeijzer (2003).

In addition, the diversity of the expression tree structure is important. we think employing a large population with fewer generations outperforms the alternative of a small population with numerous generations. That is because of the gradual loss of population diversity as the number of iterations increases, in which the localized mutation is not enough to alter the overall structure. Therefore, we use a large population size with relatively small generations to promote exploration within the population.

### C.2  COMPUTER RESOURCE

The experiments are carried out on a physical machine equipped with an Intel Core(TM) i7-8700 CPU @ 3.20GHz and a single NVIDIA RTX 3090 GPU with 24268-MB global memory.

## C.3 THE NOTATIONS USED IN THE PAPER

Table 4: Notations

| Symbol | Description |
|---|---|
| $\mathbf{s}$ | The semantics of a tree |
| $\mathbf{s}^{tr}(\mathbf{s}^j)$ | The semantics of the tree $tr$ (the $j^{th}$ tree) |
| $\mathbf{s}^c$ | The semantics of the original subtree before mutated |
| $\mathbf{s}'$ | The semantics of the generated new subtree after mutated |
| $\mathbf{s}_i$ | The $i^{th}$ dimension semantic of the semantics $\mathbf{s}$ |
| $\mathbf{st}$ | The sub-target semantics in the sub-semantic space |
| $\mathbf{st}^{tr}(\mathbf{st^j})$ | The sub-target semantics of the subtree $tr$ (the $j^{th}$ subtree) |
| $\mathbf{st}_i$ | The $i^{th}$ dimension semantic of the sub-target semantics $\mathbf{st}$ |
| $tr^i$ | The $i^{th}$ symbolic expression subtree |
| $\|\nabla T\|_{\mathcal{N},i}$ | The $i^{th}$ dimension of the normalized semantic gradient vector |
| $\|\nabla T\|_{\mathcal{N}}^j$ | The normalized semantic gradient vector of the $j^{th}$ subtree |
| $\alpha$ | The scalar factor proposed in Eq 5 |
| $\alpha_i$ | The scalar factor for the $i^{th}$ semantics |
| $k$ | The scalar value proposed in Eq 7 |
| $\eta$ | The discount factor promoting concise trees |
| $l$ | the mutated subtree size |
| $\mathbf{J}_r$ | The Jacobian matrix of the constants |
| $\mathbf{I}$ | The identity matrix |
| $\mu$ | The penalty factor in Eq 13 |
| $\beta$ | A hyperparameter in Eq 12 and Eq 13 |
| $\mathbf{w}_j$ | The $j^{th}$ constant within an expression |
| $w$ | General reference to a constant |
| $\lambda$ | A tolerance factor in Eq 8 and Eq 9 |
| $\mathcal{D}$ | A dataset of symbolic regression |
| $f(\mathbf{x};\mathbf{w})$ | The output of a symbolic expression with input $\mathbf{x}$ and the constants $\mathbf{w}$ |
| $\mathcal{L}(*)$ | A loss function with the input |
| $\mathcal{R}$ | An evaluation value of the generated subtree, which is calculated by Eq 10 |
| $\mathcal{R}^c$ | An evaluation value of the original subtree |
| $p(tr^i)$ | The probability of selecting the $i^{th}$ subtree $tr^i$ as the mutated subtree |

## D SUPPLEMENTAL BENCHMARK RESULTS

In addition to SRBench and SRSD benchmark datasets, our method have also been evaluated on several widely used datasets.

### D.1 EXPERIMENTAL RESULTS ON LIVERMORE DATASET

The range of the Livermore dataset for both training and test sets is uniformly limited to [-10, 10], with a total of 1000 sets. The training set is randomly generated within this range, while the test set adopts equal interval sampling. It should be noted that only datasets within the function set's range are selected for evaluation, given that our function set includes <+,-,*,%,sin, cos, log, exp>. Each dataset is evaluated 10 times with different seeds.

Table 5 shows the median $R^2$ performance of our method on each dataset over 10 runs, as well as the success rate under two precision settings. The $Acc_\tau$ refers to the accuracy solution rate and $\tau$ indicates the precision. It refers to successfully finding a solution when $1 - R^2 < 1.0e^{-(\tau+1)}$. The experimental results exhibit a high accuracy across most of the datasets. However, the GESR on datasets containing protection operators (Livermore-4, Livermore-12), particularly higher-order ones (Livermore-5, Livermore-12), exhibit decreased performance. The decline in fitting ability may be attributed to the interval operation protection strategy. Nevertheless, compared to traditional protection operators, this strategy can better prevent overfitting in expressions featuring numerous

constants such as geometric semantics GP. As a result, we have decided to retain the interval operation strategy.

Table 5: The median 1 - $R^2$ value and the accuracy solution rate with different tolerance on Livermore datasets.

| Dataset | Symbolic expression | $1 - R^2$ | $Acc_3$ | $Acc_{12}$ |
|---|---|---|---|---|
| **Livermore-1** | $1/3 + x_1 + sin(x_1)$ | 0 | 100% | 100% |
| **Livermore-2** | $sin(x_1^2)cos(x_1) - 2$ | 0 | 100% | 100% |
| **Livermore-3** | $sin(x_1^3)cos(x_1^2) - 1$ | 0 | 100% | 100% |
| **Livermore-4** | $log(x_1 + 1) + log(x_1^2 + x_1) + log(x_1)$ | 6.47e-3 | 20% | 0% |
| **Livermore-5** | $x_1^4 - x_1^3 + x_1^2 - x_2$ | 2.70e-12 | 100% | 30% |
| **Livermore-6** | $4x_1^4 + 3x_1^3 + 2x_1^2 + x_1$ | 1.21e-14 | 100% | 100% |
| **Livermore-9** | $\sum_{i=1}^{9} x_1^i$ | 1.99e-15 | 100% | 100% |
| **Livermore-10** | $6sin(x_1)cos(x_2)$ | 0 | 100% | 100% |
| **Livermore-11** | $(x_1^2 x_2^2)/(x_1 + x_2)$ | 0 | 100% | 100% |
| **Livermore-12** | $x_1^5/x_2^3$ | 1.23 | 0% | 0% |
| **Livermore-14** | $x_1^3 + x_1^2 + x_1 + sin(x_1) + sin(x_1^2)$ | 0 | 100% | 100% |
| **Livermore-17** | $4sin(x_1)cos(x_2)$ | 1.87e-8 | 100% | 30% |
| **Livermore-18** | $sin(x_1^2)cos(x_1) - 5$ | 0 | 100% | 100% |
| **Livermore-19** | $x_1^5 + x_1^4 + x_1^2 + x_1$ | 0 | 100% | 100% |
| **Livermore-21** | $\sum_{i=1}^{8} x_1^i$ | 0 | 100% | 100% |

## D.2 EXPERIMENTAL RESULTS ON NGUYEN DATASET

The range and number of training and test datasets for the Nguyen dataset are consistent with those of the Livermore dataset settings. As shown in Table 6, our method shows a good fitting effect in terms of accuracy when applied to the Nguyen dataset.

Table 6: The median 1 - $R^2$ value and the accuracy solution rate under different precision on Nguyen datasets.

| Dataset | Symbolic expression | $1 - R^2$ | $Acc_3$ | $Acc_{12}$ |
|---|---|---|---|---|
| **Nguyen-1** | $x_1^3 + x_1^2 + x_1$ | 0 | 100% | 100% |
| **Nguyen-2** | $x_1^4 + x_1^3 + x_1^2 + x_1$ | 0 | 100% | 100% |
| **Nguyen-3** | $x_1^5 + x_1^4 + x_1^3 + x_1^2 + x_1$ | 0 | 100% | 100% |
| **Nguyen-4** | $x_1^6 + x_1^5 + x_1^4 + x_1^3 + x_1^2 + x_1$ | 0 | 100% | 80% |
| **Nguyen-5** | $sin(x_1^2)cos(x_1) - 1$ | 0 | 100% | 100% |
| **Nguyen-6** | $sin(x_1) + sin(x_1 + x_1^2)$ | 0 | 100% | 100% |
| **Nguyen-7** | $log(x_1 + 1) + log(x_1^2 + 1)$ | 3.29e-7 | 100% | 20% |
| **Nguyen-9** | $sin(x_1) + sin(x_2^2)$ | 0 | 100% | 100% |
| **Nguyen-10** | $sin(x_1)cos(x_2)$ | 0 | 100% | 100% |
| **Nguyen-12** | $x_1^4 - x_1^3 - 0.5x_2^2 + x_2$ | 0 | 100% | 100% |

## D.3 FURTHER EXPERIMENTAL RESULTS ON SRSD BENCHMARK

In this section, the accuracy solution rate and symbolic solution rate of the GESR on the SRSD benchmark(with and without dummy variables) are presented.

The results show that the GESR achieves significantly better performance in terms of accuracy solution rate on both types of SRSD datasets, while is also competitive in terms of symbolic solution rate. However, It must be pointed out that, compared to the symbolic solution rate, the proposed GESR is better at numerical fitting under a limited model complexity.

Table 7: SRSD: Accuracy solution rate($R^2 > 0.999$) on scientific discovery datasets with different complexity.

| Group | gplearn | AFP | AIF | DSR | E2E | uDSR | PySR | GESR |
|---|---|---|---|---|---|---|---|---|
| **Easy** | 6.67% | 20.0% | 33.3% | 63.3% | 26.7% | 100.0% | 66.7% | 100.0% |
| **Medium** | 7.50% | 2.50% | 5.0% | 45.0% | 17.5% | 75.0% | 45.0% | 92.5% |
| **Hard** | 2.00% | 4.00% | 6.00% | 28.0% | 14.0% | 20.0% | 38.0% | 64.0% |

Table 8: SRSD: Symbolic solution rate on scientific discovery datasets with different complexity.

| Group | gplearn | AFP | AIF | DSR | E2E | uDSR | PySR | GESR |
|---|---|---|---|---|---|---|---|---|
| **Easy** | 6.67% | 20.0% | 30.0% | 46.7% | 0.00% | 50.0% | 60.0% | 53.3% |
| **Medium** | 0.00% | 2.50% | 2.50% | 10.0% | 0.00% | 17.5% | 30.0% | 40.0% |
| **Hard** | 0.00% | 0.00% | 2.00% | 2.00% | 0.00% | 4.00% | 4.00% | 6.0% |

Table 9: SRSD +Dummy Variables: Symbolic solution rate on scientific discovery datasets with different complexity.

| Group | gplearn | AFP | AIF | DSR | E2E | uDSR | PySR | GESR |
|---|---|---|---|---|---|---|---|---|
| **Easy** | 0.00% | 16.7% | 0.00% | 10.0% | 0.00% | 10.0% | 20.0% | 3.3% |
| **Medium** | 0.00% | 0.00% | 0.00% | 0.00% | 0.00% | 7.50% | 5.00% | 10.0% |
| **Hard** | 0.00% | 0.00% | 0.00% | 2.00% | 0.00% | 0.00% | 0.00% | 2.00% |

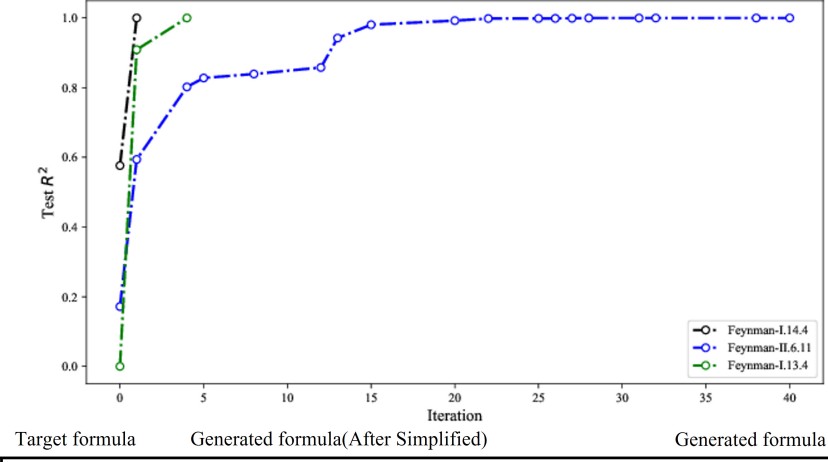

Figure 6: Qualitative analysis of the equation mutation process.

### D.4 Qualitative analysis of the GESR

The process of searching for symbolic solutions and accuracy solutions are further investigated respectively. The $R^2$-Iteration convergence curve and the final formulas are illustrated in the Figure 6. We can find that, the idea of providing weights for each combined subtree and optimizing using the LM algorithm to adjust the global tree structure works well(Many redundant subtree weights are optimized to 0, like Feynman-I.13.4). Another interesting phenomenon is that, although our method tends to perform the numerical fitting when the formula is complicated, $R^2$ can also be reduced to nearly 0 even if the symbolic solution is not found.

Table 10: Examples of the evolution formulas.

| Dataset | Target formula | Simplied formula | Generated formula | 1-$R^2$ |
|---|---|---|---|---|
| **Accuracy Solution(1-$R^2$ < 1e-15)** | | | | |
| **i.30.5** | $x0/(x1*sin(x2))$ | $1.0 * x0 * (-x0 + x1)/(x1 ** 2 * sin(x2))$ | $((x1 - x0) * 1.002764 * 1.017231)/sin(x2)/((x1/x0 * -0.0 + (x1 + x2) * 0.0 + x1)/(x0/x1/(x1/x1) * 1.0 + x0 * x2 * -0.0)) * 0.980352 + ((x0/x1/(x2 - x0) * -143652773275.769 + x0/x1/(x2 * x0) * 4.213271) * -0.003478 + log(log(x0))/(x1/x2 * 2.6e - 05 * x2 * x2) * 0.0) * -0.0$ | 0 |
| **ii.34.11** | $x0 * x1 * x2/(2 * x3)$ | $0.5*x0*x1*x2/x3+ 1.31 * x0 * x2 ** 2 + x0 * x2 - x2 ** 2 * x3 + 0.346 * x2 ** 2 + 0.346 * x2$ | $x2 * x2 * (x0 - x3) + x0 * x2 + x2 * (((x1/x1 + x1 + x2) - (x3 - x0)/(x3/x1)) * 0.345614 + (x1/x3 + x2 + x2) * x0 * 0.154386)$ | 0 |
| **ii.2.42** | $x0 * x3 * (x1 - x2)/x4$ | $-1.01*x0*x3*(x1- x2) * (0.002 * x4 - 1)/x4$ | $x3*(((x2/x4*-0.030838+x2* x2*-0.0)-((x1-x2)*-5.5e- 05 + x1/x4 * -0.030838)) * ((x0*0.001796+x1*0.0)*x4- x0)*-32.427845+((x4+x3)* (x2 - x1) + cos((x4 + x4))) * ((x3-x0)*2e-06+(x0*x1- x4/x3) * -0.0) + (x2 - x3) * (x4+x4)*x0*-1e-06+(x3- x4 + x0) * (x4 * 57732.18215 + x1 * 52.941879) * 0.0)$ | 0 |
| **iii.17.37** | $x0*(x1*cos(x2)+ 1)$ | $x0 - x1 * (x0 - x1 * *2)$ | $x0 - (x0 - x1 * x1) * x1$ | 0 |
| **Symbolic Solution** | | | | |

| i.27.6 | $1/(x1/x2+1/x0)$ | $1.0 * x0 * x2/(x0 * x1 + x2)$ | $(x2 + (x2 * x1 - (x0 - x2)) * 0.0 + x2 * x0 * -61.494388 * -0.0)/(x2/x0 * 1.213848 + (x2 + x0 * x2) * 0.0 + x1 * 0.002891 * 419.809517 + (x1 - x2) * 0.0) * (sin(cos((0.882887 * 0.882103))) * 1.858405 + 1.0 * 0.0) + ((x0 * -389.595871 + x2 * -300.836646 + (x0 - x2) * -3.8092 + 1.0 * -0.70011 + x0/x1 * -7.939006) * 0.0 + x0 * x2 * log(x1) * log((x1 * x2)) * (x0 * x0 * 0.0 + (x2 - x0) * -0.0) + log(x1)/log(x1)/(x1/x0 - (x1 + x2)) * 0.0) * 2.889718$ | 0 |
| i.14.3 | $9.81 * x0 * x1$ | $9.81 * x0 * x1$ | $(x1 * x0)/(x1/x1) * x1/x0/(x1/x0) * 9.80665 + log((x0/x0 * log(x1))) * ((x1 + x0) - (x0 - x0))/(x1 - x0 - log(x1)) * 0.0$ | 0 |
| ii.3.24 | $0.08 * x0/x1 * *2$ | $0.08 * x0/x1 * *2$ | $x0/(x1 * x1 * x0/x0) * 0.079577 + ((x0 * x0 + x0/x0)/((x1 + x0) * x0 * x1) - (x0/x1 + x0/x0) * (x0 * x0 + x0/x1)) * -0.0$ | 0 |
| i.14.4 | $0.5 * x0 * x1 * *2$ | $0.5 * x0 * x1 * *2$ | $x0 * x0 - x0 * x0 + x1 * x1 * x0 * (0.5 + (x1 - x0 + x0 * x1 + x0/x0 - (x0 - x0)) * 0.0)$ | 0 |
| i.26.2 | $sin(x0)/sin(x1)$ | $1.0 * sin(x0)/sin(x1))$ | $x0/x0 * (x1 + x0) * x0/((x0/x0 * sin(x1))/(sin(x0)/(x0 + x0)) * 0.5 + log((log(x0)/(x0 - x0))) * -0.0)/(x0 + x1)$ | 0 |
| **Approximate Solution(1-$R^2$ < 1e-3)** | | | | |
| bonus.18 | $(x0 * *2 * x2 * *2 * x3 * *2 * (x3 * x4/x5 + 1) + x1 * *2)/(2 * x0)$ | $-0.25 * x3 * x4 * (x0 - x5) + 0.5 * x1 * *2/x0$ | $(x3 * x4 * (x0 - x5) - (x1 + x1) * x1/x0) * -0.25 + ((x3 + x4) - x3 * x3 + x5/x0/(x0 * x2)) * (x5/x0/(x5 - x2) - (x0 + x2) * (x1 - x5)) * -0.0$ | 4.88e-14 |

| i.15.3x | $x0 - x1 * x2$ | $1.0*x0 - x2*(0.998* x1 + 5.29)$ | $((x2 - x0 - x1 * x2) * -7.42079873031672e + 17 + x2*x1*6462889543579853.0 + x1/x2 * 471097.535999) * (x1/x0 * -0.0 + x0/x2 * -0.0 + cos(log(x1)) * -0.035472) * -0.0 + ((x2 + x2 + x1 * x2) * 6.008895307088127e + 17 + x2 * x1 * log(x1) * 0.129268 * -2.2813953662454227e + 17) * (cos(cos(log(x1))) * -0.0 + (x1/x2 * 0.0 + 0.09026) * 0.0) + x2 * ((x1 * -0.999754 + -5.295591 + x2 - x2 + x0/x2) * 0.998293 + (x1 * 1.417872 + x2 * 346065318.206917) * (x1 * x1 - x2/x0) * -0.0) + (-3.76412 + x0 * 1.250469) * (x1 + x1) * (x2 + x1) * 0.0 + (x1 * -0.0 + 94.293878) * x0 * 1.6e - 05 + ((x0 * 0.628168 + x1 * -0.0 + sin(x1) * sin(x2)) - cos((x0 * x0))/(x1/x1 * (x2 + x0))) * 0.000848$ | 3.14e-8 |
|---|---|---|---|---|
| i.34.10 | $x0/(1 - 0.333e - 8 * x1)$ | $1.0 * x0 + 0.00101 * x0/x1 + 0.499 * log(x1)$ | $(((x1 * -13653.999224 + x0 * 1.258642) - ((x1 + x0) - x1 * x0)) * -0.999914 + (x0 * -164713.383376 + 118328192256260.03 + log(x1) * x1 * x0) * -5e - 06) * -0.0 + (log(x1) * x1 * x0 * 0.156508 + (x1 * x0 - x1 * x1) * -1.331564) * ((x1 * x1 + x0 + x0) * 1.012645 + x0 * x1 * 2e - 06 + (x1 - x0) * -16.983238) * 0.0 + (((x0 + x0 + log(x1)) * -0.998994 + (x0/x1 + x0) * -0.002012) * -0.499967 + (x0/x1 + x0 - x1) * 6.5e - 05 + (log(x1) - (x0 + x0)) * (log(x1) + x1/x1) * 0.0) - x0/((x0/x1 + x0 + x1) * 1.000529 + x1 * x1 * (x0 + x1) * -0.0) * (x1 * x1 * (x0 + x1) * (-0.0 + x1 * -0.0) + x1 * x0 * (x1 + x1) * 0.0 + x0 * x1 * 0.0 + x1 * x1 * 0.0 + x1 * x1 * (x0 + x1) * 1.0 * -0.0)$ | 3e-16 |

| | | | | |
|---|---|---|---|---|
| **i.39.11** | $x1*x2/(x0-1)$ | $x2*(-1.06*x0*x1*(0.058*x0-1.0)+(-0.457*x0*x1-0.961*x1-48.3*x2)*sin(log(x0))*cos(2*x0/x1))/sin(log(x0))$ | $(x2*x1*((x0*x0*0.057574+1.000004*4e-06)-(x0*1.0034*0.996614+x0*x0*5e-06)))/sin(log(x0))*-1.061088+((x2*-1.000007+x0*-0.0)*-96.710375+x2*7.701144*-12.428074)/(-0.026219+x0*0.000476+5e-06+cos(x2)*8e-06+log(x0)*2e-06)*(x1*x0*0.456581+(x0+x1)*-0.000264+x1*-0.039793+x2*47.286156+x1/x2*-0.0+x1+x2)*(cos((x0/x1+x0/x1))*0.025936+(cos(log(sin(x0)))+cos((0.99726+x0*0.001665)))*(7.4e-05+(cos(x0)+x0)*3e-05))$ | 4e-16 |
| **Special Condition** | | | | |
| **iii.7.38** | $1.9ed+34*x0*x1$ | $zoo*x0*x1+1$ | $(x0*x1*(x1-x0))/((x1-x0)*x0*x1)-(x1/x1*(x0-x0)-(x0*x1)/(cos((x1*x0*(x0-x0)))*0.0+(x0-x1)*x1*x0*(x1/x1-(x1+x1))*0.0))$ | 0 |
| **i.48.2** | $8.99e+16*x0$ | $zoo*x0*(2.28e+21*x0-0.036*log(x0))*(zoo*x0+1)$ | $(((x1*-1.0+x0*6.2866856149886544e+22)*1.0+x1-log(x0))*(0.036195+cos(log(x1))*-0.0)+(log((x1+x1))/(log(x1)-(x0+x0))+log((x1/x0))+x1*x0*(x1+x0))*-2e-06+(x1*x1*1643323978166933.5+x1*-0.005684*7.673404519338928e+24+x1*0.998321*-2.4933595845987697e+22+x1/x0*-0.0+x0/x1/(x0+x1)*-1.7018808659799349e+68+x1/x0/(x1+x0)*11.321839)*0.0)*(x0+x0*-3.10136811332914e+38*-0.0+(x0*x0)/(x1-x1)*-1.060634018085812e+22)/(x1*(x1*0.0*5558069507319.527+x1/x0*0.0+(x0+x1)*x1/x0*-0.0))/(x1/x1/(x1*x1)*1.0*0.0+cos((log(x1)-cos(x0)))*x1*-1.005799*0.0+(-1.007637+x1/x1/(x1*x1)*-430533244100.9507)*0.0)$ | 7.14e-11 |

| | | | | |
|---|---|---|---|---|
| **ii.21.32** | $8.99e + 9 * x0/x1$ | $-8.99e + 9 * x0/(x1 * (0.454 * cos((x1 + x2)/x2) + zoo))$ | $x0/((x1 * -0.633341 * -3.281687 + x1 * x2 * -0.0) * (0.999981 + x1 * -0.0) * 0.506802 + (x1 * x2 + log(x0)) * 0.0 + (x1 * 1.95036 + x2 * 0.0) * 0.00026)/(log((log(x2) * (x1 - x1))) * -0.045066 + cos(((x2 + x1)/x2)) * -0.453885 + (x1/x1 * (x2 - x1) + x1 + x0 + x1 - x2) * (1.0 * 0.0 + (x1 - x0) * -0.0)) * 1.0 * 9471418140.76377$ | 5.06e-14 |
| **i.37.4** | $x0 + x1 + 2 * sqrt(x0 * x1) * cos(x2)$ | $0.224 * (0.303 * x0 * x1 + 4.42 * x0 + 4.42 * x1 + 0.00598 * cos(x2)) * (-0.00598 * x0**2 + 0.169 * cos(x0) - 0.00598 * cos(x1) + 0.00598 * cos(x1 - x2) + 0.848) + (0.00598 * x0/x1 + (-1.13 * x0 + 1.13 * x1 + 31.9) * (0.187 * x0 + 0.187 * x1 + 0.00101) + 0.381 * log(x0 * x1) - 0.013 * log(x0 * *2 - x0 + x1) + 3.05 * log(log(x0 + x1)) + 0.00598 * x1/x0) * sin((x0 + x1) * cos(x0**2) * cos(x2))$ | $(cos(x0) * -0.169021 + -1.853386 * 0.457336 + ((x0 * x0 + cos(x1)) - cos((x2 - x1))) * 0.00576) * (((x2 - x0) * (x2 - x2) + x2 - x0 - (x2 + x1)) * -4.422398 + (cos(x2) * 11.632138 + x1 * x0 * 549.516861) * 0.000551) * -0.223539 + sin((cos((x0 * x0)) * cos(x2) * (x1 + x0))) * ((-31.893814 + (x1 - x0) * -1.130586) * (-0.001229 + (x0 + x1) * -0.186696) + log((x1 - x0 + x0 * x0)) * -0.012644 + log((x0 * x1)) * 0.380954 + x0/x1 * 0.00594 + x1/x0 * 0.005865 + log(log((x0 + x1))) * 3.054007)$ | 1.51e-4 |

| bonus.7 | $3.0e + 8 * sqrt(-x1/x2 **2)$ | $(7.83e+10*x0* x1*x2 + x0*x2* (-9.23e+33*x0 - 1.21e+9*x1 + 2.66e+8)*log(x2) - 8.18e+9*x1)/(x2* log(x2))$ | $((x2/x1 - (x2 + x2)) * (x1 - x1 + x1*x0)*0.0 + cos(log((x2/x0)))* -0.0 + (x0+x1)*sin(x1)* -0.0 + (cos((log(x2) + x0*x2))* 1.243243 + (x2+x0+x2*x1)* (x0-x1+x0+x1)*0.049346)* 0.0) - ((((x1*x0)/log(x2)* 1878.48632 + x1*x0* -28.949997 + x2*x1* -0.0)* -41687892.707671 + ((x0/x1* 8.906147285880547e + 53 + x1*x1* -6.044006062559795e + 27) - x0/x0/(x0 + x1))* (x0*x2*0.0 + (cos(x1) + cos(x1))*(-0.0 + x0* 0.0))) - (x1/x2/log(x2)* -8177765343.762911 + x1/x2* 1.8666636319898988e + 26* -0.0 + x0*246026712.333492 + x2*0.0 + x0*x2*0.0 + x0* x0* -9.232410317538276e + 33 + x0*20340985.054632 + x2*0.0 + (1.256028 + x0* -2.3055888900197576e + 25 + x1*0.748889 + 0.013151)* (1.0*0.0 + cos(x1)*0.0)))$ | 1.67e-4 |

Besides, in Table 10, we also present some of the other generated formulas along with their simplified forms, including the accuracy solutions, symbolic solutions, approximate solutions, and special conditions on the SRSD datasets. In most instances, the generated formulas exhibit good interpretability after being simplified using the sympy library. However, when the necessary operators for obtaining the target formula are lacking (such as 'sqrt'), the complexity of the simplified formula may increase to approximate the target formula as closely as possible. Additionally, it is noteworthy that some generated formulas, after simplification using the sympy library, yield 'zoo' values despite having an R2 value of 1. Such a situation primarily arises due to the protected division operator. The sub-expression $f(x)$ as a denominator with zero-value is replaced with $\sqrt{1 + f(x)^2}$ during the computation.

## D.5 SCALABILITY TO DATA SIZE

Since the semantic dimension is associated with the training set size, there may be a spatial sparsity problem when dealing with a large-scale dataset. It is also the reason why Eq.4 is used in the geometric search operator to quickly approximate the semantic vector to the sub-target semantics first. To further analyze the specific performance of the GESR under different data scales, the black-box dataset in SRBench according to the data scale is re-partitioned, and the $Avg.R^2$ of the top-ranked methods is statistically compared in SRBench, in which the results show the competitiveness of the GESR.

Table 11: The Avg.$R^2$ performance on different data scales.

| | $0\sim100$ | $100\sim500$ | $500\sim1000$ | $1000\sim5000$ | $5000\sim10000$ | $10000\sim$ |
|---|---|---|---|---|---|---|
| GESR | **0.665** | **0.872** | **0.891** | 0.618 | 0.857 | **0.632** |
| SBP-GP | **0.665** | 0.857 | 0.876 | 0.608 | 0.858 | 0.630 |
| Operon | 0.580 | 0.870 | 0.890 | **0.623** | **0.861** | 0.629 |
| FEAT | 0.664 | 0.867 | 0.886 | 0.619 | 0.857 | 0.632 |
| DSR | 0.584 | 0.605 | 0.559 | 0.500 | 0.699 | 0.386 |

## D.6 DETAILED RESULTS ON STROGATZ DATASETS

In order to better observe the specific performance of our method dealing with scientific discovery data sets, we further evaluate our method on the Strogatz datasets. Table 12 shows the median value $(1\text{-}R^2)$ for 10 independent runs of our method on the Strogatz datasets, in which $1\text{-}R^2$ is labeled 0 when $R^2$ achieves the accuracy to tolerance 15. Figure 7 shows the solution rate of our method on the Strogatz series datasets, in which solution rate means the rate of finding the symbolic solution.

Table 12: The median $1 - R^2$ on Strogatz datasets.

| Dataset | Operon | SBP-GP | AIFeynman | MRGP | DSR | GESR |
|---|---|---|---|---|---|---|
| **Strogatz_bacres1** | 3.56e-7 | 3.51e-6 | 6.19e-3 | 7.02e-5 | 1.14e-1 | 3.66e-6 |
| **Strogatz_bacres2** | 2.67e-8 | 2.19e-9 | 2.91e-4 | 2.59e-5 | 1.17e-1 | 2.43e-5 |
| **Strogatz_barmag1** | 2.76e-6 | 1.76e-9 | 0.75 | 7.33e-5 | 1.73e-1 | 0 |
| **Strogatz_barmag2** | 1.70e-7 | 0 | 0 | 4.91e-5 | 1.26e-1 | 0 |
| **Strogatz_glider1** | 1.31e-12 | 0 | 4.99e-3 | 3.68e-5 | 1.14e-1 | 0 |
| **Strogatz_glider2** | 1.85e-7 | 1.83e-5 | 0 | 4.55e-5 | 0 | 0 |
| **Strogatz_lv1** | 2.24e-11 | 0.16 | 9.73e-1 | 2.78e-1 | 1.93e0 | 0 |
| **Strogatz_lv2** | 1.40e-11 | 1.66e-4 | 0 | 2.95e-2 | 5.55e-1 | 0 |
| **Strogatz_predprey1** | 3.25e-7 | 1.02e-3 | 1.01e0 | 2.83e-4 | 8.15e-2 | 1.11e-3 |
| **Strogatz_predprey2** | 7.68e-6 | 2.78e-4 | 5.11e-3 | 8.04e-5 | 1.71e-1 | 1.08e-4 |
| **Strogatz_shearflow1** | 8.75e-3 | 6.33e-3 | 0 | 3.43e-4 | 0 | 0 |
| **Strogatz_shearflow2** | 5.11e-9 | 4.18e-11 | 1.05e0 | 3.98e-5 | 3.02e-1 | 0 |
| **Strogatz_vdp1** | 3.58e-9 | 6e-5 | 1.03e0 | 4.91e-5 | 2.47e-1 | 0 |
| **Strogatz_vdp2** | 4.99e-15 | 0 | 0 | 2.94e-5 | 1.10e-1 | 0 |

As can be seen from Table 12, we can achieve fitting effect on most datasets of Strogatz datasets. For a few formulas that do not fit perfectly, there are also a good numerical fitting effect. The underlying formulas for the four data sets that failed to fit are: $y = 20 - x_1 - \frac{x_1 \cdot x_2}{(1+0.5 \cdot x_1^2)}$, $y = 10 - x_1 \cdot x_2/(1+0.5 \cdot x_1^2)$, $y = x_1 \cdot (4 - x_1 - \frac{x_2}{1+x_1})$, $y = x_2 \cdot (\frac{x_1}{x_1+1} - 0.075 \cdot x_2)$. A common feature can be found from these formulas, that is, they all have protective division. This may means that interval operations to some extent prevent the generated formulas from generating extreme values, but also limit the ability of GESR to search for some formulas. Thus, the exploration of protection operators is also a possible direction for our future research.

Figure 7 shows the success rate of our method in finding the perfect formula. It can be observed that, compared with the advantages of accuracy fitting, our method does not show great advantages in terms of the solution rate. Although the purpose of our method is on numerical fitting, it is undeniable that the high-intensity numerical fitting ability of geometric semantic variation also limits the ability to find a perfect formula. When fitting the underlying expression with high complexity, there may be multiple mismatched parts in an expression that are expected to be replaced. Once the subexpression tree where all mismatched parts are located cannot be found and replaced correctly in the process of geometric semantic variation, Then, due to the semantic approximation property, the mismatched but

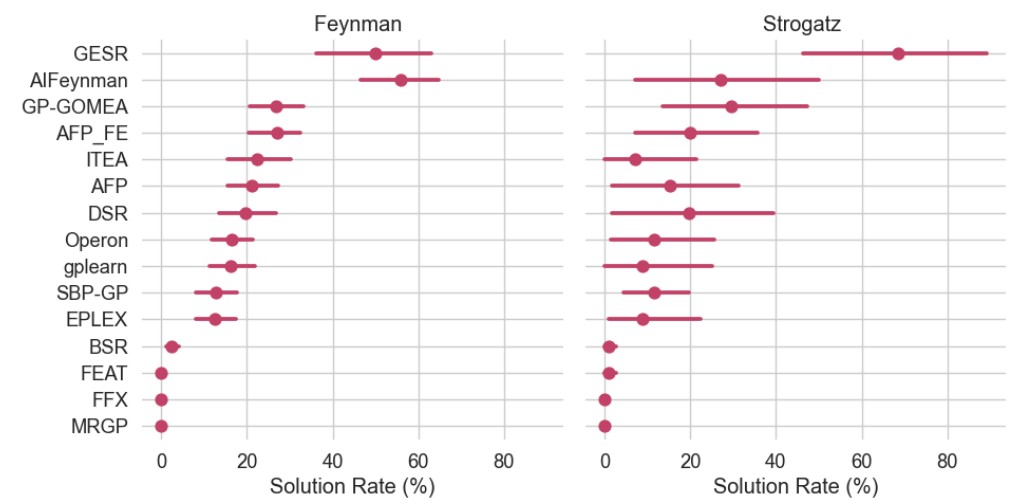

Figure 7: Solution rates on Feynman datasets(left) and Strogatz datasets(right) of SRBench benchmark.

semantically more similar sub-expression tree may be chosen for replacement. This will cause the symbolic regression process to be transformed into a numerical fitting process rather than a search for a perfect formula, which is also an issue worth studying and deepening in the future.

