# OpenReview forum: "GESR: A Geometric Evolution Model for Symbolic Regression"
_ICLR.cc/2025/Conference — ICLR 2025 Conference Withdrawn Submission_

### Official Review · Reviewer_qUzG · 2024-10-31

**Soundness:** 2
**Presentation:** 2
**Contribution:** 2
**Rating:** 5
**Confidence:** 5

**Summary:**

In this paper, the authors propose a concept of semantic gradient to help genetic programming evolve better.

**Strengths:**

The strength of this paper is its strong performance on SRBench.

**Weaknesses:**

The weakness of this paper is that the contribution is unclear. The paper needs significant improvement to clarify the advantages of the proposed method over traditional semantic backpropagation.

**Questions:**

Here are some questions that need to be addressed:
1. What is the difference between the proposed method and traditional semantic backpropagation with linear scaling [1]?
2. Using the Levenberg-Marquardt algorithm to optimize constants is not novel in the symbolic regression domain [2]; it should not be claimed as a contribution.
3. The definition of $M$ on line 324 is confusing, as both $i$ and $j$ are mixed. Please ensure consistent notation.
4. In Equation (6), it seems the proposed method linearly combines two subtrees with weights. Why doesn’t this method suffer from the issue of exponential growth [3]? Please explain.
5. Equations (8) and (9) only eliminate some candidates from the candidate set, and the remaining candidates are evaluated. How does the proposed method ensure a fair comparison of computational complexity with baseline methods?
6. A related question is that training time is not shown in Figure 4. Without a comparison of training times, it is hard to know whether the proposed method genuinely improves existing methods or simply increases computational budget to achieve better results.
7. Pseudocode 1 should be included in the main paper instead of the supplementary material. Without the pseudocode, it is difficult to understand the proposed method.
8. The final section, "Conclusion," is missing a section number. Please fix this issue.
9. There should generally be a space between the text and references. Please add spaces where needed.
10. It is unclear why both $\alpha$ and $k$ are needed in Equation (6). $\alpha$ alone should be sufficient, and least squares can determine the optimal $\alpha$.

[1]. Virgolin, Marco, Tanja Alderliesten, and Peter AN Bosman. "Linear scaling with and within semantic backpropagation-based genetic programming for symbolic regression." Proceedings of the Genetic and Evolutionary Computation Conference. 2019.
[2]. Kommenda, Michael, et al. "Parameter identification for symbolic regression using nonlinear least squares." Genetic Programming and Evolvable Machines 21.3 (2020): 471-501.
[3]. Martins, Joao Francisco BS, et al. "Solving the exponential growth of symbolic regression trees in geometric semantic genetic programming." Proceedings of the Genetic and Evolutionary Computation Conference. 2018.

---

> ### Author Response · Authors · 2024-11-19
> **Rebuttal by Authors**
>
> Clarification and explanation for Weakness:
>
> 1.	I think it is necessary to emphasize the contribution of this paper, although we have claimed our contribution multiple times in the original manuscript. One of the important contributions of this paper is discovering the inconsistency of the approximation process in sub-semantic space and target semantic space and attempting to solve it by proposing the concept of semantic gradient. Secondly, considering the excessive sparsity in the semantic space, we propose a new geometric semantic method, which is used to quickly approximate the target semantics in the sparse semantic space. Third, we present the concept of using a continuous optimization algorithm to adjust the tree structure of the semantic-based method. Although you think this is not a contribution, we stand by our views. For the detailed reasons, please see reply 3.
>
> Answer for Questions:
>
> 2.  Regarding the difference between our geometric semantic method and the method in [1], with a careful reading, you will find that the two methods are totally different in both strategy and the final performance. By the way, we have already compared it with our method in both the R2 and expression size in the original manuscript, in which the method in [1] is named SBP-GP in the comparison. In addition, we have also made the relevant ablation experiment in the original manuscript, in which the performance drops when we replace our method with the method in [1].
>
> 3.	The main reason why we regard this as a contribution is that in this paper we present the concept of using continuous optimization algorithms to adjust the tree structure, while the purpose of using LM in [1] [2] is to just optimize constants. And since we assign a constant (weight) to each combination subtree, there is a natural characteristic for our method to use a continuous optimization algorithm to adjust the overall tree structure. Its effect can be seen in the qualitative analysis (Appendix D.4): the weight parameters can be optimized to zero to eliminate useless subtrees.
>
> 4.	Thanks for your suggestion, we have revised it in the updated manuscript.
>
> 5.	Our geometric semantic method is a replacement of the subtree by using the backpropagation with the semantic gradient, assisted with a strict subtree screening mechanism (Equations 8 ~ 10), rather than directly combining the two whole trees. By the way, we have also made a limitation on the model size.
>
> 6.	We have supplemented the runtime performance, which is shown in Figure 4 of the updated manuscript. It is worth noting that due to the differences in the fields to which the baseline method belongs (evolutionary computation / deep reinforcement learning / deep learning, etc.), a completely fair comparison is difficult to achieve.
>
> 7.	Thanks for your suggestion, what you say makes sense. We have supplemented the runtime comparison in the updated manuscript. From Figure 4 we can find that, our runtime is still in a reasonable range, and there is even an advantage over the methods with top R2 performance.
>
> 8.	Thanks for your suggestion. Actually, the main process shown in the pseudocode has already been drawn and explained in Figure 1. The original intention is to hope that the main paper can focus on describing the several most important modules. To make it easier for the readers to understand the whole process, we have improved the schematic (Figure 1) and corresponding caption. Lines 80 ~ 88 (Introduction), and lines 165 ~ 175 have also been revised in the updated manuscript.
>
> 9. Thanks for your suggestion. The manuscript has been updated with the corresponding revised.
>
> 10. Thanks for your suggestion. The manuscript has been updated with the corresponding revised.
>
> 11.  $\alpha$ and $k$ correspond to two steps: migration and approximation. These two steps are indispensable. The $\alpha$ values assigned to the two subtrees are not equal ($\alpha_1$ and $\alpha_2$), which means that they cannot be solved by the least squares method. If there is only one subtree, the $\alpha$ value can only be used for linear scaling and cannot make a geometric approximation.

---

> ### Comment · Reviewer_qUzG · 2024-11-22
>
> Thanks. However, three major concerns still need to be addressed:
>
> 1. What is the inconsistency between the approximation process in the sub-semantic space and the target semantic space? Section 3.2 attempts to explain this, but it is unclear. Please provide a concrete example for clarification.
>
>     What does "excessive sparsity in the semantic space" mean? Why does the proposed method effectively address this issue?
>
> 2. Regarding the current loss function, as shown in Equation (12), it is unclear how some weight parameters can be optimized to zero. Typically, optimizing coefficients toward zero requires L1 regularization.
>
> 3. Linear regression can be used to determine the optimal coefficients of multiple variables. The response provided is not convincing.

---

> ### Author Response · Authors · 2024-11-23
> **Thanks for the continued disscusion**
>
> > Reply to Comment 1:
>
> Thank you for your comment. We will further elaborate this concept. For the distance between the semantics $\mathbf {s}$ of subtree  $tr$  and the sub-target semantics  $\mathbf {st}$  and the difference $diff^s_i$: $|\mathbf {s}_i - \mathbf {st}_i|$ on each dimension $i$ in the sub-semantic space, due to the presence of features ($\mathbf {x}$ in Figure 2) in the computation path when sub-semantics propagate to the root node ($w\cdot2\mathbf{x}$ in Figure 2), **each dimension undergoes a varying degree of scaling**. Specifically, the difference $diff^s_i$ on each dimension $i$  is transformed at the root node to $diff_i=|(w\cdot2\mathbf{x}_i)\cdot(\mathbf{s}_i - \mathbf{st}_i)|$. This results in significant deviations in the approximation process, thereby severely impacting the approximation effectiveness within the sub-semantics space, especially in cases with complex computation paths. Hope such a description can help you.
>
> - What does "excessive sparsity in the semantic space" mean? Why does the proposed method effectively address this issue?
>
> The dimension of the semantic space is equal to the training set size, which is quite large (The corresponding description can be found in line 160 and lines 252~253 in the manuscript). The proposed two-step geometric semantic method makes it possible to use a geometric method to quickly approximate the target semantics directly in the semantic space.
>
> > Reply to Comment 2:
>
> Actually, we aim to transform the process of adjusting the expression into an n-dimensional continuous optimization problem (where n represents the number of weights), by assigning weights to each mutated subtree every time in the geometric semantic mutation. For instance, there is a generated formula $f$ and there exists an optimal weight set $\mathbf{\theta}^*=\\{\theta_1, \theta_2, …, \theta_n\\}$ such that $f=f^*$ (where $f^*$ denotes the target expression), in which its subset $\theta’=\\{\theta \in \theta^* | \theta=0\\}$ is non-empty. Then the Levenberg-Marquardt (LM) algorithm can be utilized for continuous optimization of the expression to solve for the optimal $\{\theta^*\}$ set, achieving the effect of removing uninformative subtrees. Its effect is also shown in the Figure 6 and Table 9. Besides, it is noteworthy that the LM algorithm in this paper is used to optimize the weight parameters of each subtree to attain a global weight balance and facilitate adjusting the expression structure, which differs from the optimization of constants conducted in other papers.
>
> > Reply to Comment 3:
>
> Firstly, the reviewer should also consider the computational burden. Since there are many candidates for each subtree, it is expensive to perform matrix inversion or iterative calculation.
>
> Secondly, multiple linear regression requires that there should be no multicollinearity among the variables, which cannot be guaranteed during the process of subtree combination.

---

> ### Comment · Reviewer_qUzG · 2024-11-24
>
> Thanks. The idea of using gradient information to guide semantic mutation is good. However, I still have two comments corresponding to the first two questions:
>
> 1. To solve the inconsistency between the approximation process in the sub-semantic space and the target semantic space, gradient information is proposed as shown in Equation (3). However, the gradient should be with respect to the loss $\mathcal{L}$, instead of the function $f$, to directly reflect its influence on the loss [1].
>
> 2. The response about using the least square method (Levenberg-Marquardt algorithm) without $L_1$ regularization for sparse optimization is not convincing. Without $L_1$, redundant variables are difficult to eliminate. This can be simply verified by the following example:
> ```python
> import numpy as np
> from sklearn.linear_model import Ridge, Lasso
>
> X = np.random.randn(50, 20)  # 50 samples, 20 features
> true_weights = np.zeros(20)
> true_weights[:10] = np.random.randn(10)  # Only the first 10 weights are non-zero
> y = X @ true_weights
>
> ridge = Ridge(alpha=0.1)
> ridge.fit(X, y)
> print(ridge.coef_)
>
> lasso = Lasso(alpha=0.1)
> lasso.fit(X, y)
> print(lasso.coef_)
> ```
>
> Moreover, I have two additional comments that could further improve the quality of this paper:
>
> 3. On page 6, $t$ and $m$ are two hyperparameters that determine the number of combinations for trials. Please provide the specific values used in this paper.
>
> 4. Please provide statistical test results to validate the effectiveness of the proposed method. Specifically, the proposed method should demonstrate statistically significant improvements over SBP-GP.
>
> [1]. Graff, M., Graff-Guerrero, A., & Cerda-Jacobo, J. (2014). Semantic crossover based on the partial derivative error. In Genetic Programming: 17th European Conference, EuroGP 2014, Granada, Spain, April 23-25, 2014, Revised Selected Papers 17 (pp. 37-47). Springer Berlin Heidelberg.

---

> ### Author Response · Authors · 2024-11-25
> **Thanks for the continued disscusion**
>
> Thanks for your continued comments.
> > Reply to Comment 1:
>
> Sincerely, thank you for your further suggestions. However, the purpose of the proposed semantic gradient is to indicate the changing rate in each dimension to address the problem we have proposed.  The role of the loss gradient in the method mentioned by the reviewer differs from the role of the semantic gradient in our method. The paper mentioned by the reviewer uses the gradient information to aid in constant optimization and to select the crossing point that roughly conforms to the approximate direction of the loss decrease (just judge the positive or negative) of the candidate subtree. Instead, our method is to use the semantic gradient to indicate the changing rate in each dimension thereby determining the importance of each dimension(scaling degree). Given that the loss function is highly relevant to the current tree structure and the original subtree awaiting mutation, it seems not suitable for our method.
>
> Please feel free to correct me if there are any mistakes.
>
> > Reply to Comment 2
>
> Thanks for your comment. I misunderstood your question before. The usage of L2 instead of L1 is to better smooth the curve from the consideration of the capability to resist the noise and there is also a precision setting for the generated expression. We have updated the description to avoid misunderstanding:
>
> 'by setting the corresponding subtree's weight coefficient to 0.0' -> 'by setting the corresponding subtree's weight coefficient to 0.0 when the weight coefficient is approximated to nearly zero($<$1e-6) in our method).'
>
> > Reply to Comment 3
>
> Thanks for your comment. We have provide these two hyper-parameters in the updated manuscript.
>
> > Reply to Comment 4
>
> Actually, the effectiveness has already been demonstrated in the ablation analysis. The 'GESR-mutation' is to replace our geometric semantic approximation method with the SBP-GP operator.

---

> > ### Comment · Reviewer_qUzG · 2024-11-26
> >
> > For the statistical test, I want to clarify that it refers to using the Wilcoxon signed-rank test to evaluate the statistical significance between algorithms, as shown in [1]. This test is widely used in the symbolic regression domain to compare different algorithms [1-3].
> >
> > [1]. Graff, M., Graff-Guerrero, A., & Cerda-Jacobo, J. (2014). Semantic crossover based on the partial derivative error. In Genetic Programming: 17th European Conference, EuroGP 2014, Granada, Spain, April 23-25, 2014, Revised Selected Papers 17 (pp. 37-47). Springer Berlin Heidelberg.
> >
> > [2]. Virgolin, M., Alderliesten, T., & Bosman, P. A. (2019, July). Linear scaling with and within semantic backpropagation-based genetic programming for symbolic regression. In Proceedings of the genetic and evolutionary computation conference (pp. 1084-1092).
> >
> > [3]. La Cava, W., Burlacu, B., Virgolin, M., Kommenda, M., Orzechowski, P., de França, F. O., ... & Moore, J. H. (2021). Contemporary symbolic regression methods and their relative performance. Advances in neural information processing systems, 2021(DB1), 1.

---

> ### Author Response · Authors · 2024-11-26
> **Thanks for the continued disscusion**
>
> Thanks for your comment.
>
> To reply to your question, now the scipy.stats.ranksums function is used on the median experiment results of the Friedman datasets to test whether there are statistically significant differences (p < 0.01) with and without the inclusion of the three main contributions (weight optimization module, geometric semantic method, semantic gradient). Compared with the GESR-baseline, GESR-opt (without weight optimization module), GESR-gradient (without semantic gradient module), and GESR-mutation (replacing the geometric semantic method with SBP-GP) yields p values below 1e-4, 1e-5, and 1e-4, respectively. The p values by using scipy.stats.wilcoxon are below 1e-06, 1e-07, 1e-06 respectively.

---

> > ### Comment · Reviewer_qUzG · 2024-12-03
> >
> > Based on the significance test, I believe this paper represents a meaningful improvement for SBP-GP. Accordingly, I have raised the score to 5. However, the paper requires further clarification to present the algorithm more clearly.
> >
> > Additionally, the paper could consider comparing the heuristic approach to pruning subtrees with coefficients less than $1e^{-6}$ against the use of L1 regularization. In fact, with L2 regularization, all coefficients may shrink to very small values, potentially eliminating even important subtrees. This is why neural network-based methods often utilize L1 or L0.5 regularization [1].
> >
> > [1]. Kim, S., Lu, P. Y., Mukherjee, S., Gilbert, M., Jing, L., Čeperić, V., & Soljačić, M. (2020). Integration of neural network-based symbolic regression in deep learning for scientific discovery. IEEE Transactions on Neural Networks and Learning Systems, 32(9), 4166-4177.

---

> ### Author Response · Authors · 2024-12-03
> **Thanks for your continued discussion**
>
> Thanks for your suggestion. Since the deadline for manuscript revisions has passed, we will not make changes now. The following reply is just for discussion purposes.
>
> The L1 regularization may indeed be better at feature selection, while the L2 is better at balancing the weights to resist the noise. However, one point that should be noticed is that the usage of L2 in our method is based on the weights generated by the geometric semantic method, while the network-based method is mostly based on the normalized weights. The conditions are different between our method and the network-based symbolic regression like EQL. From Table 10 it can be observed that there are also some weights with large values. However, the L1 regularization may indeed be a better choice if we focus on the accuracy solution rate and compromise on the capability of resisting noise interference. Thanks for your suggestion, we will consider replacing L2 with L1.

---

### Official Review · Reviewer_ZPa5 · 2024-10-31

**Soundness:** 3
**Presentation:** 2
**Contribution:** 2
**Rating:** 5
**Confidence:** 3

**Summary:**

The authors present a new method for symbolic regression based on geometric semantics. They introduce the Semantic Gradient, a novel approach that guides adjustments to an expression tree by indicating the direction and magnitude of changes required within each subtree. This approach enables more targeted and effective mutations during the evolutionary process. Additionally, the authors introduce other key steps, such as a variance-based constraint and a novel evaluation function, to avoid local overfitting and enhance exploration. The constants are then optimized using the Levenberg-Marquardt (LM) method with L2 regularization to improve robustness.
In their experiments, they demonstrate that this new methods achieve state-of-the-art performance (in terms of R2>0.999) on both the SRBenchmark and the SRSD+ Dataset.

**Strengths:**

The authors demonstrate that their method achieves state-of-the-art performance across a wide range of benchmarks.
They have released the code with their submission (which however could not be run due to the absence of the requirements.txt file)

**Weaknesses:**

The methodology is unfortunately difficult to follow and understand. A clear example guiding the reader through the key steps and illustrating how these steps work together (either in the main paper or in an appendix) would have been valuable.

For the SRSD+ dataset, there appears to be a significant performance improvement over other baselines; however, some important methods are missing from the comparison (though they are included in SRBench). The authors should also consider testing competitive baseline methods such as Operaon and SBP-GP on this benchmark.

An analysis of computational requirements is missing. It is unclear how long the algorithm takes to fit an equation and how this compares to other baselines. Ideally, it would be useful to see the performance of this novel method against some of the other baselines with an equivalent computational budget.

**Questions:**

In your experiments, did you use the same computational budget as the baselines? If not, could you conduct an evaluation where the same compute time is allocated to your approach, Operaon, and SBP-GP, and analyze the results?

How does your method compare with others in terms of interpretability? Are the resulting formulas human-readable, or do they tend to be extremely long? It might be interesting, for example, to have the symbolic form of all your expressions on the Feynman dataset.

A similar approach, SBP-GP, seems almost equivalent to your method in terms of performance in Figure 4 and Table 9, yet you outperform it on Feynmann and Strogatz (Figure 5). Do you have an explanation for this?

---

> ### Author Response · Authors · 2024-11-19
> **Rebuttal by Authors**
>
> Clarification and explanation for Weakness:
>
> 1.	Thanks for your suggestion. We have provided the pseudocode (in Appendix A) and made a corresponding description. Furthermore, to make it easier for readers to understand, now we have improved the schematic (Figure 1) and corresponding caption. Lines 80 ~ 88 (Introduction) and lines 165 ~ 175 have also been revised in the updated manuscript.
>
> 2.	The datasets of the SRSD benchmark are based on the physics formulas from the Feynman Lectures Series, which is also included in SRBench (The experiment results can be seen in Figure 5). The difference just lies in the rigor and scientificity of the data range in SRSD and the addition of dummy values. By the way, while [1] has also shown the SOTA performance on the scientific discovery datasets of SRBench benchmark in their paper and we have got the better performance in both benchmarks (in which the [1] is included in SRSD), it is enough to demonstrate the effectiveness of our method.
>
> 3.	We have presented the runtime performance in the updated manuscript. Our runtime is in a reasonable range, and there is even an advantage over the methods with top R2 performance in Figure 4.
>
> Answer for Questions:
>
> 4.	We have supplemented the runtime statistics in Figure 4 of the updated manuscript. Actually, the time constraints in SRBench are relatively relaxed (48 hours for black-box problems and 8 hours for ground-truth problems, in which our method is far below the limit). This is because the goal of scientific discovery should be to find a better solution in a certain time range, rather than the time competition. By the way, from the additional runtime statistics, we can find that there is even an advantage over the methods with top R2 performance in Figure 4.
>
> 5.	We have done the qualitative analysis in Appendix D.4 in the original manuscript. After the generated formula is simplified, the readability can be greatly improved. In addition, it should be noted that the constants of the generated formula in Fig.3 are also consistent with the relevant constants of the target formula (despite some constant values looking quite large).
>
> 6.	Figure 4 is based on black-box datasets (also known as real-world datasets, such as the red-white wine dataset or stock prediction dataset). Actually, there is a lot of interference in the black-box dataset with all types of noise and dummy variables, and their internal logic is often more complex and the feature size is also much higher than the scientific discovery datasets. We can notice that based on the black-box datasets, the performance of the several top methods is relatively close (Figure 4), while there is a gap between the methods in the scientific datasets (Figure 5). Table 9 is a further statistic results based on the black-box datasets.
>
> [1] Landajuela M, Lee C S, Yang J, et al. A unified framework for deep symbolic regression[J]. Advances in Neural Information Processing Systems, 2022, 35: 33985-33998.

---

> > ### Comment · Reviewer_ZPa5 · 2024-11-21
> > **Response by Reviewer**
> >
> > Weakness 2: It's still unclear why different baselines are used in different benchmarks. In its current form, the presentation suffers significantly. I strongly recommend aligning the benchmarks to ensure consistency in the baseline methods considered, especially for the top performing ones. Strong baselines should be included across both datasets. Additionally, ensure consistency between the text and captions. For example, in Table 1, the term “SRSD+ Dummy Variables” is used, whereas in the text and your response, you refer to “SRSD.” Why are different terms being used?
> >
> > Weakness 3 / Question 4: It is interesting to see that your method achieves a reasonable runtime. Thank you for including this information.
> >
> > Question 5: In its current form, Appendix D.4 includes only three equations, which is quite limited, and it is unclear how these equations were selected. I suggest including all AiFeynman equations, either in the appendix or in an additional file, and placing greater emphasis on interpretability.
> >
> > Question 6: I recommend incorporating these intuitions into the main paper.

---

> > > ### Author Response · Authors · 2024-11-23
> > > **Thank you for the continued discussion**
> > >
> > > Thanks for your comment.
> > >
> > > > Reply to the Comment 1
> > >
> > > Thanks for your suggestion. However, it is worth noting that the baseline methods are selected by the corresponding benchmark papers rather than intentionally screened by us. The original intention of using the additional SRSD benchmark is to increase the persuasiveness of the SOTA performance since many works only use one of them (SRBench or RSRD, with the corresponding baseline methods) as a benchmark. There will be lots of work to align these two benchmarks and there is really no need to do this (Actually, most of the baseline methods in the SRSD benchmark are also selected based on their symbolic solution rate performance on SRBench, while some other new methods like uDSR [1], E2E [2] and PySR [3] are the recent popular studies. The corresponding description can also be found in the SRSD benchmark paper). If it is really strange, how about only using one benchmark, like many other papers do?
> > >
> > > > Reply to the Comment 3
> > >
> > > Thanks for your suggestion sincerely. Given that there are a total of 120 formulas, presenting all of them in the manuscript could result in an excessively lengthy and cumbersome table, even spanning potentially over ten pages. However, to reply to your question, in Table 9 of the updated manuscript, we have categorized and presented many generated formulas based on the types: accuracy solution, symbolic solution, approximate solution, and special conditions. The presented Table endeavors to cover the conditions associated with the generated formulas as much as possible. It includes the generated formulas derived from datasets of various difficulties (easy, medium, hard) and encompasses multiple special cases, such as high formula complexity generated formulas and those that yield zoo values after simplification using the sympy library. Corresponding analyses are also provided in lines 1275~1284:
> > >
> > > "Besides, in Table 9, we also present some of the other generated formulas along with their simplified forms, including the accuracy solutions, symbolic solutions, approximate solutions, and special conditions on the SRSD datasets. In most instances, the generated formulas exhibit good interpretability after being simplified using the sympy library. However, when the necessary operators for obtaining the target formula are lacking (such as 'sqrt'), the complexity of the simplified formula may increase to approximate the target formula as closely as possible. Additionally, it is noteworthy that some generated formulas, after simplification using the sympy library, yield ‘zoo’ values despite having an R2 value of 1. Such a situation primarily arises due to the protected division operator. The sub-expression \(f(x)\) as a denominator with zero-value is replaced with \(\sqrt{1 + f(x)^2}\) during the computation.”
> > >
> > >
> > > > Reply to the Comment 4
> > >
> > > Thanks for your suggestion.  Actually, most of these description has appear in the SRBench benchmark paper. However, to reply to your comment, we add some relevant descriptions in Section 4.3 of the updated manuscript:
> > >
> > > " Compared to the black-box datasets with the unknown underlying data generated function containing substantial irregular noise and dummy variables, there a more pronounced distinctions among the baseline methods for the capability to solve the scientific discovery datasets."
> > >
> > > [1] Landajuela M, Lee C S, Yang J, et al. A unified framework for deep symbolic regression[J]. Advances in Neural Information Processing Systems, 2022, 35: 33985-33998.
> > > [2] Pierre-Alexandre Kamienny, Stéphane d’Ascoli, Guillaume Lample, and François Charton. Endto-end Symbolic Regression with Transformers. In Advances in Neural Information Processing Systems, volume 35, pages 10269–10281, 2022.
> > > [3] Miles Cranmer. Interpretable Machine Learning for Science with PySR and SymbolicRegression.jl. arXiv preprint arXiv:2305.01582, 2023.

---

> > > > ### Comment · Reviewer_ZPa5 · 2024-11-23
> > > > **Response to the Author**
> > > >
> > > > I understand and agree that running a large number of baselines goes beyond the scope of this work. However, in the text you have written, you state:
> > > >
> > > > “Our method achieved accuracy solution rates (R² > 0.999) of 100.0%, 87.5%, and 58% on the easy, medium, and hard datasets, respectively. This means that our method improves by 23.3%, 42.5%, and 36% over the second-ranked method.”
> > > >
> > > > Please correct me if I’m mistaken, but while this claim is valid, it’s possible that one of the methods shown in Figure 4 might outperform the second-ranked method, which could nullify your claim. I suggest ensuring that the strongest current baselines (two or three should suffice) are run and reported for each dataset. Specifically, you should include uDSR, SBP-GP, and Operon  for both methods.
> > > >
> > > > Comment 3:
> > > >
> > > > This looks good to me. Just ensure that the remaining expressions are easily accessible on your GitHub repository; a .csv file would suffice.
> > > >
> > > > Comment 4:
> > > >
> > > > Looks good.

---

> > > > > ### Author Response · Authors · 2024-11-23
> > > > > **Thank you for the continued discussion**
> > > > >
> > > > > Thanks for your prompt comments sincerely. Actually, the uDSR [1] has reported their sota performance on the scientific discovery datasets of the SRBench benchmark in their paper, that is also one of the reasons we further use the RSRD benchmark to evaluate our methods. But if you insist that it is necessary to further reproduce some methods on both datasets (it is indeed possible that Operon or SBP-GP performs better than uDSR on SRSD, though unlikely since the uDSR has reported the sota performance on the SRBench), we will try to reproduce the Operon or SBP-GP on the RSRD benchmark and report it in the later updated manuscript.
> > > > >
> > > > > [1] Landajuela M, Lee C S, Yang J, et al. A unified framework for deep symbolic regression[J]. Advances in Neural Information Processing Systems, 2022, 35: 33985-33998.

---

> > > > > > ### Author Response · Authors · 2024-11-24
> > > > > > **Reply to Comment 3**
> > > > > >
> > > > > > Thanks for your suggestion. The remaining expressions with R2 have been uploaded to GitHub, which is named 'SRSD_generated_formulas.txt'. The reviewer can also find it in the anonymous link.

---

### Official Review · Reviewer_m6md · 2024-11-02

**Soundness:** 3
**Presentation:** 2
**Contribution:** 3
**Rating:** 5
**Confidence:** 3

**Summary:**

This paper propose GESR, a symbolic regression method Leveraging geometric semantics. the process of symbolic regression in GESR is transformed into an approximation to an unimodal target in n-dimensional  topological space.  A new geometric search operator is proposed to improve the search efficiency of the algorithm. The experimental results show that GESR has good performance.

**Strengths:**

1. Introducing SGP into symbolic regression. To a certain extent, it can indeed improve the traditional GP algorithm mutation and crossover without guidance, improving the overall efficiency of the algorithm.

2. From the experimental results of the paper, GESR obtained very good results.

**Weaknesses:**

1. Why is the maximum noise level chosen in the article only 0.01? I remember that the noise level of the SRBench dataset was added to 0.1. Please explain why the algorithm's performance at 0.1 noise level was not tested.
2. The experiment is not complete, there is no inference time (for the same expression, the time spent by different algorithms to obtain the same index, which is related to the operation efficiency of the algorithm), expression complexity evaluation (such as the number of symbols of the expression. Because it is meaningless to obtain a good goodness-of-fit with a very complicated expression), which is extremely important to prove the performance of the algorithm.
3. The baselines selected in this article are quite old. Why did the author choose these baselines? I suggest that the author compare with the latest algorithms.  Such as SNIP(https://doi.org/10.48550/arXiv.2310.02227) and so on. And, if we have DSR, why not compare it with the stronger DSO(NGGP):https://doi.org/10.48550/arXiv.2111.00053?  These strong baselines are more reflective of the comprehensive performance of GESR.

**Questions:**

1.  I hope the author highlights and emphasizes your motivation more clearly in the introduction section.
2. The algorithm needs to be improved in the methods section, which as a whole is hard to follow and took a long time to understand. I also want to reorganize the language. For example, how GESR guides mutation and crossover.
3. Below Equation 1. Semantic genetic programming (GP) -- > Semantic genetic programming (SGP).
4. The resolution of the pictures in Figure 4 and Figure 5 seems to be insufficient, and they are blurred when enlarged

---

> ### Author Response · Authors · 2024-11-19
> **Rebuttal by Authors**
>
> Thanks for your comments.
>
> Clarification and explanation for Weakness:
>
> 1.	Thanks for your comment. For why there is no experiment on the 0.1 dataset, with a careful observation of the original SRBench paper you can find that most of the methods have a very significant R2 performance degradation under this level of Gaussian noise (The accuracy solution rate of many methods in the Strogatz datasets have even been reduced to 0). Considering that the main purpose of this paper is not to enhance the robustness to resist strong noise, I think that 0.001 and 0.01 are sufficient to reflect the adaptability of our method to Gaussian noise; The strong robustness to anti-noise is not the main research scope of this paper.
> 2.	Thanks for your suggestion. The runtime performance has been supplemented in Figure 4 of the updated manuscript and the expression complexity evaluation has already been given as the model size in Figure 4 of the original manuscript. We can observe that the model size is much smaller than the second-rank method. Our runtime performance is also in a reasonable range and there is even an advantage over the methods with top R2 performance in Figure 4.
> 3.	We mainly use the two most popular benchmarks: SRBench and SRSD. SRSD benchmark already contains newer methods such as uDSR. In addition, most of the recently emerging methods, including uDSR and SNIP, still use the same SRBench benchmark as a baseline, which means that we can easily compare with the experimental data presented in their paper. From the performance shown in their papers, we can find that there still have been advantages to our method on R2 performance. By the way, now we also try to compare our method with several GSGP methods (SLGP[3], ADGSGP[2]), uDSR[1], and TPSR[4] in Feynman datasets. (Since SPL, Mundhenk, and uDSR are all integrated into one Dso package. We reproduce the uDSR algorithm based on the parameters proposed in [1].)
>
>     ||Avg.$Acc_3$|Avg. $Acc_{12}$ |Avg. inference time|
>     |--|--|--|--|
>     |GESR	|**0.95**	|**0.75**|1399.12|
>     |uDSR	|0.77	|0.68	|5131.13|
>     |TPSR	|0.82	|0.52	|**228**|
>     |ADGSGP	|0.67|0.42|40236.62|
>     |SLGP	|0.64|0.23|11925.44|
>
> Answer for Questions:
>
> 4.	Thanks for your suggestion. The main motivations and contributions are as follows: 1) On the basis of finding the inconsistency of the approximation process in sub-semantic space and target semantic space, we attempt to solve it by proposing the concept of semantic gradient; 2) Considering the excessive sparsity in the semantic space, we propose a new geometric semantic method, which is used to quickly approximate the target semantics in the sparse semantic space; 3) In view of the lack of continuous optimization ability in the current symbolic regression, we present the concept of using continuous optimization algorithm to adjust the tree structure generated by the semantic-based method. The relevant description has been updated in lines 81 ~ 87 (Introduction) and lines 165 ~ 174 (preliminary) of the updated manuscript.
>
> 5.	The proposed geometric semantic method is a mutation method, which has been described in detail in Section 3.3 of the original manuscript and has also been presented in Figure 1. The overall process can also be followed through Figure. 1.
>
> 6.	Thanks for your suggestion. The typo has been revised in the updated manuscript.
>
> 7.	Thanks for your suggestion. We have improved the clarity of the figures.
>
>
> [1] Landajuela M, Lee C S, Yang J, et al. A unified framework for deep symbolic regression[J]. Advances in Neural Information Processing Systems, 2022, 35: 33985-33998.
> [2] Chen Q, Xue B, Zhang M. Improving generalization of genetic programming for symbolic regression with angle-driven geometric semantic operators[J]. IEEE Transactions on Evolutionary Computation, 2018, 23(3): 488-502.
> [3] Huang Z, Mei Y, Zhong J. Semantic linear genetic programming for symbolic regression[J]. IEEE Transactions on Cybernetics, 2022, 54(2): 1321-1334.
> [4] Shojaee P, Meidani K, Barati Farimani A, et al. Transformer-based planning for symbolic regression[J]. Advances in Neural Information Processing Systems, 2023, 36: 45907-45919.

---

> > ### Comment · Reviewer_m6md · 2024-11-25
> > **For the author's reply**
> >
> > Thank you very much for your earnest reply. I would like to make the following comments regarding your reply.
> >
> >
> > 1, I did not see SNIP and DSO as baselines in the paper.
> >
> > 2. **Model Size** and **Training Time (Hr)** in Figure 4 are mostly just a point without confidence intervals (horizontal bars), while $R^2$ does. I feel this result is unreasonable.
> >
> > All in all, considering your responses to other reviewers, I decided to keep the score as it is. Thank you very much for your reply

---

> > > ### Author Response · Authors · 2024-11-25
> > > **Thanks for your reply**
> > >
> > > Thanks for your comments. It is ok to keep the score and I also respect your decision.
> > >
> > > However, to avoid possible misunderstandings, one point that must be clarified is that there exists confidence intervals for the Model Size and Training Time of all methods. Just they are much shorter compared to the confidence intervals of R2 ( the performance is relatively more stable compared to R2) for most baseline methods. The reviewer can also observe the same phenomenon in the original SRBench paper.

---

### Official Review · Reviewer_SUeL · 2024-11-04

**Soundness:** 2
**Presentation:** 1
**Contribution:** 1
**Rating:** 3
**Confidence:** 4

**Summary:**

This paper proposes the geometric evolution symbolic regression (GESR) algorithm which introduces new concepts of semantic gradients and semantic mutation to the existing geometric semantic genetic programming.

**Strengths:**

Sufficient relevant literature is cited in related works.

**Weaknesses:**

The clarity of the paper could be improved with clearer self-contained definitions of the terminologies used. For example, in the introduction, the paper should have a self-contained description of what ‘semantic space’ is in the context of this paper, to avoid potential misinterpretation that it is a novel concept introduced. “Underlying n-dimensional topological space” is not a sufficient description given that “n” is not even defined.

The paper should also be clearer on what is novel. For instance, the section on constant optimization might be misinterpreted as a novelty when it has been used in several prior works such as [1] and [2].

Selection of SR algorithms used for comparison are not consistent across datasets. In SRSD datasets, more recent works such as E2E and uDSR are evaluated, but these are missing in the SRBench datasets. Likewise, non-SR based algorithms are evaluated in the SRBench datasets, but not for SRSD datasets.

This paper lacks comparison against more recent SR algorithms such as [3]. This is important because the main strength of the algorithm introduced in this paper is in its prediction performance, which should include more recent works which also claim to have improved prediction performance. Otherwise, the paper should explain the criteria for selecting comparison algorithms and include a strong justification if recent methods are excluded.

The paper lacks important comparison against geometric semantic evolution algorithms. Examples such as GSGP (and others mentioned in line 105 to 106 of the paper) are cited in the second paragraph of “Related Works”, but never compared to in the evaluation.

The algorithm proposed seem impractical for real-world usage since the equation size is large (above a size of 100 on average), especially in comparison to existing symbolic regression methods, as shown in Figure 4. Since the paper introduces SR as a machine learning method with the key strength of interpretability, the size of the equations discovered by the proposed method does not seem consistent with the interpretable aspect.

The paper should also tune the hyperparameters of the benchmark SR methods to encourage larger model size to determine if higher R2 test performances could be achieved by trading-off equation size. Currently, it is unclear if GESR simply performs better on R2 test because it allows for a larger equation size.

Since the semantic approach does not scale as well with the size of the data, a runtime analysis of the methods (i.e., training time) should be conducted and reported. Currently, it is unclear to the reader whether the comparison against existing SR algorithms was done under fair settings.

[1] Kommenda, M., Burlacu, B., Kronberger, G., & Affenzeller, M. (2020). Parameter identification for symbolic regression using nonlinear least squares. Genetic Programming and Evolvable Machines, 21(3), 471-501.

[2] De Melo, V. V., Fowler, B., & Banzhaf, W. (2015, November). Evaluating methods for constant optimization of symbolic regression benchmark problems. In 2015 Brazilian conference on intelligent systems (BRACIS) (pp. 25-30). IEEE.

[3] Shojaee, P., Meidani, K., Barati Farimani, A., & Reddy, C. (2023). Transformer-based planning for symbolic regression. Advances in Neural Information Processing Systems, 36, 45907-45919.

**Questions:**

In the abstract, methodology and conclusion, why is the Levenberg-Marquardt algorithm framed as a key module that is proposed/introduced by the paper when it has already been utilized in many prior works like [1] and [2]?

Can the paper use a consistent set of SR benchmark algorithms across the various datasets and include comparison against existing geometric semantic genetic programming algorithms (see works cited in lines 105 to 106 of the paper)?

The paper states that "The proposed algorithm significantly improves the fitting performance of symbolic expression while
effectively tackling concerns associated with tree bloating", yet the results (specifically Figure 4), do not provide evidence for this.

[1] Kommenda, M., Burlacu, B., Kronberger, G., & Affenzeller, M. (2020). Parameter identification for symbolic regression using nonlinear least squares. Genetic Programming and Evolvable Machines, 21(3), 471-501.

[2] De Melo, V. V., Fowler, B., & Banzhaf, W. (2015, November). Evaluating methods for constant optimization of symbolic regression benchmark problems. In 2015 Brazilian conference on intelligent systems (BRACIS) (pp. 25-30). IEEE.

---

> ### Author Response · Authors · 2024-11-19
> **Rebuttal by Authors [1]**
>
> Clarification and explanation for Weakness:
>
> 1.	The semantic genetic programming and the semantic space have been explained in the preliminary, and the related semantic works are presented in the related work, I don’t think it will cause misunderstanding to the readers. About the description of 'n', thanks for your suggestion. Although we have claimed that the space dimension is equal to the training set size (line 249), it may indeed confuse readers. We have supplemented a relevant description in the updated manuscript.
>
> 2.	I think it is necessary to emphasize the contribution of this paper again, although we have claimed our contributions multiple times in the original manuscript. One of the important contributions of this paper is discovering the inconsistency of the approximation process in sub-semantic space and target semantic space and attempting to solve it by proposing the concept of semantic gradient. Secondly, considering the excessive sparsity in the semantic space, we propose a new geometric semantic method, which is used to quickly approximate the target semantics in the sparse semantic space. Third, we present the concept of using a continuous optimization algorithm to adjust the tree structure of the semantic-based method. Despite you think this is not a contribution, we stand by our views.
> The main reason why we regard this as a contribution is that in this paper we present the concept of using continuous optimization algorithms to adjust the tree structure, while the purpose of using LM in the papers you list is to just optimize constants. And since we assign a constant (weight) to each combination subtree, there is a natural characteristic for our method to be able to use a continuous optimization algorithm to adjust the overall tree structure. Its effect can be seen in the qualitative analysis (Appendix D.4): The weight parameters can be optimized to zero to eliminate useless subtrees.
>
> 3.	I have to say, while [1] has also shown the SOTA performance on the scientific discovery datasets of SRBench benchmark in their paper and we have got the better performance in both benchmarks (in which the [1] is included in SRSD), it seems no need to evaluate the baseline methods of SRBench again on the SRSD benchmark.
>
> 4.	We have tried to compare our method with several GSGP methods (SLGP[3], ADGSGP[2]), uDSR[1], and TPSR[4] in Feynman datasets. By the way, since SPL, Mundhenk, and uDSR are all integrated into one dso package. We reproduce the uDSR algorithm based on the parameters proposed in [1].
>
>     ||Avg.$Acc_3$|Avg. $Acc_{12}$ |Avg. inference time|
>     |--|--|--|--|
>     |GESR	|**0.95**	|**0.75**|1399.12|
>     |uDSR	|0.77	|0.68	|5131.13|
>     |TPSR	|0.82	|0.52	|**228**|
>     |ADGSGP	|0.67|0.42|40236.62|
>     |SLGP	|0.64|0.23|11925.44|
>
> 5.	It should be noted that SBP-GP is a novel geometric semantic method (also cited in the related work). One of the purposes of this paper is to achieve better r2 test performance with the restricted model size constraints. The results in Figure 4 and Figure 5 have demonstrated the effect of the proposed method (no matter in R2, accuracy solution rate, and expression size). And we have also compared the proposed method with some new GSGP methods additionally, which can be seen in reply 4.
>
> 6.	SBP-GP is also a semantic method, and our method can maintain better R2 performance while ensuring a significantly smaller size, which I think is enough to illustrate the effectiveness of the method.
>
> 7.	Just like we have explained in reply 6， our model size has been significantly reduced compared to the second-ranked method in the SRBench benchmark. In addition, if you carefully read the SRBench benchmark paper[5], you can find that there is no strict limit on the model size, and the experimental results of the baseline methods in the original paper are obtained through the half-grid search with multiple sets of hyperparameters. The model size shown in Figure 4 of our original manuscript also ranges from $10$~$10^6$.
>
> 8.	In terms of the scalability of our semantic method with the dataset size, we have specifically made statistics on the effectiveness of the methods under different dataset sizes in Appendix D.5 of the original manuscript. As for the training time, the main reason we didn’t show the runtime before is based on the consideration of the differences in the fields to which the baseline method belongs and the differences in the usage of devices. For example, for the methods based on the generative model, GPU is used to accelerate this process, while some traditional methods run on a single CPU thread. However, we must acknowledge that the presentation of training time can serve as a reference. Therefore, we are willing to show the current average training time, and it has been presented in Figure 4 of the updated manuscript.

---

> > ### Author Response · Authors · 2024-11-19
> > **Rebuttal by Authors [2]**
> >
> > Answer for Questions:
> >
> > 9.	Please see reply 2. We present the concept of using a continuous optimization algorithm to adjust the tree structure generated by the semantic-based method while the methods you mentioned aim to optimize the constants.
> >
> > 10.	Please see replies 3 and 4. By the way, the baseline methods are selected by the corresponding benchmark paper rather than intentionally screened by us. Besides, the datasets in the SRSD benchmark are based on the physics formulas from the Feynman Lectures Series, which are also included in SRBench (The experiment results can be seen in Figure 5). The difference lies in the rigor and scientificity of the data range and the addition of dummy values in SRSD. Since we have shown better performance in both benchmarks, it is enough to demonstrate the effectiveness of our method.
> >
> > 11.	The corresponding explanation has been presented in replies 6 and 7.
> >
> > [1] Landajuela M, Lee C S, Yang J, et al. A unified framework for deep symbolic regression[J]. Advances in Neural Information Processing Systems, 2022, 35: 33985-33998.
> > [2] Chen Q, Xue B, Zhang M. Improving generalization of genetic programming for symbolic regression with angle-driven geometric semantic operators[J]. IEEE Transactions on Evolutionary Computation, 2018, 23(3): 488-502.
> > [3] Huang Z, Mei Y, Zhong J. Semantic linear genetic programming for symbolic regression[J]. IEEE Transactions on Cybernetics, 2022, 54(2): 1321-1334.
> > [4] Shojaee P, Meidani K, Barati Farimani A, et al. Transformer-based planning for symbolic regression[J]. Advances in Neural Information Processing Systems, 2023, 36: 45907-45919.
> > [5] La Cava W, Burlacu B, Virgolin M, et al. Contemporary symbolic regression methods and their relative performance[J]. Advances in neural information processing systems, 2021, 2021(DB1): 1.

---

> > > ### Comment · Reviewer_SUeL · 2024-11-19
> > >
> > > > Bullet Point 9
> > >
> > > Please see my response to bullet point 2 of your reply above.
> > >
> > > > Bullet Point 10
> > >
> > > I understand that SRBench and SRSD selected the set of benchmark methods. However, quite some time has passed since these papers were released, especially SRBench, so it is unfair to exclude evaluation on state-of-the-art methods. Also, a key strength of GESR is in its prediction performance, which other papers like TPSR claim to have too, so it would be expected that a comparison is made to these papers. In fact, in the TPSR paper, E2E results are included in their SRBench dataset results. For this reason, I do not agree that the current results are sufficient to conclude that the effectiveness of GESR has been shown.
> > >
> > > > Bullet Point 11
> > >
> > > Please see my response to bullet point 6 and 7 of your reply above.

---

> > > > ### Author Response · Authors · 2024-11-19
> > > > **Thank you for the continued discussion**
> > > >
> > > > Thanks for your prompt comments sincerely.
> > > >
> > > > > Reply to Comment 1:
> > > >
> > > > Thanks for your suggestion. We now remove the 'n-dimensional' in line 67 and move it to line 159 for the first mention and description.
> > > >
> > > > > Reply to Comment 2:
> > > >
> > > > It is worth noting that this one contribution focuses on presenting the concept of using a continuous optimization algorithm to make **adjustments of the model structure**, while the other papers including the [1] and [2] you have mentioned before focus on the parameter optimization (there is also a relevant description in [1]: 'During local search the model structure E remains fixed for a model M, while the parameter vector is subject to optimization', which can be found in the bottom of page 478). Let's use equation(6) in the paper [1] which you mentioned as an example. While the model structure is $M_{E,\theta} = \frac{(\theta_1 (x_1 - \theta_2) * (x_3 - \theta_3))}{x_2^2 (x_1 - \theta_4)}$ in that paper, our model structure may be $M_{E,\theta}' = \frac{(\theta_1^* * (x_1 - \theta_2) * (x_3 - \theta_3) + \theta_2^* * M_{E2, \theta} + ... + \theta_3^* * 1)}{(\theta_4^* * (x_2^2 (x_1 - \theta_4)) + \theta_5^* * M_{E4, \theta})}$. Then, what the meaning of 'the adjustment of model structure' is that while the target expression becomes $1 / x_2^2 (x_1 - \theta_4)$, it is also possible for our model to fit the expression by optimizing $\theta_1^*$, $\theta_2^*$, and $\theta_4^*$ to zero, which is unable for $M_{E,\theta}$. Thus, what we want to emphasize is the idea of using a continuous optimization algorithm to adjust the model structure, rather than only optimize the constants. By the way, to avoid misinterpretation, we will cite the relevant papers in Section 3.5 later to emphasize the difference. Thanks for your suggestion and hope my reply can solve your question.
> > > >
> > > > > Reply to Comment 3:
> > > >
> > > >  What I mean is that since the uDSR has shown SOTA performance in the same scientific discovery datasets of SRBench in their own paper(actually, many recent papers use SRBench as the main baseline), we can easily judge the performance by directly comparing the results presented in both papers, in which our method seems to perform better in terms of R2 performance in the scientific discovery datasets of SRBench. Additionally, the SRSD benchmark is also based on the Feynman datasets, and SRBench includes both black-box datasets and Feynman datasets. While we have presented the significant advantage on R2 performance compared to the second-ranked method in the Feynman datasets of SRBench, there seems no need to transfer the baseline methods of SRBench to SRSD again. (Actually, the selection strategy of these baseline methods in the SRSD paper is also based on the performance of the methods in SRBench).
> > > >
> > > > > Reply to Comment 7:
> > > >
> > > > Please see reply 3.
> > > >
> > > > > Reply to Comment 10:
> > > >
> > > > Your comment makes sense, and we have already presented the comparison with TPSR. By the way, whether the performance is SOTA can be easily judged since recent methods mostly use the same SRBench benchmark as a baseline, which is also one role for a benchmark. For the effectiveness of the contributions, I think whether a contribution is effective should depend on the novelty of a concept and the relative improvement compared to the same type of algorithms. The SOTA performance is just to demonstrate the power and the potential of a method.

---

> > > > > ### Comment · Reviewer_SUeL · 2024-11-21
> > > > >
> > > > > > On the Authors' Reply to Comment 2:
> > > > >
> > > > > This still does not address my concern. The concern is that the line
> > > > >
> > > > > "Then, three key modules are proposed to enhance the approximation: (1) ...; (2) ...; (3) the Levenberg-Marquardt algorithm with L2 regularization, used for the adjustment of the expression structures and the optimization of constants."
> > > > >
> > > > > is potentially misleading. What you intend to communicate is what you have written in the reply. But the concern that I have (which I see that another reviewer has mentioned too), is that your listed contributions do not match what you intend to communicate. Currently, the paper still seems to claim contribution for using Levenberg-Marquardt algorithm with L2 regularization for the optimization of constants in SR algorithms, which has already been done in a relatively well-known prior work.
> > > > >
> > > > > The line
> > > > >
> > > > > "(3) We introduce the concept of using a continuous optimization algorithm to assist in adjusting the expression structure."
> > > > >
> > > > > is also potentially misleading in the same sense.
> > > > >
> > > > > To reiterate, the contributions listed do not accurately match what is done in the paper. The current contributions listed, in my opinion, can be easily misinterpreted to overlap with other prior well-known work in SR.
> > > > >
> > > > > > On the Authors' Reply to Comment 3, 7, 10:
> > > > >
> > > > > The authors' claim " we have already presented the comparison with TPSR.". The authors have uploaded a revision to the paper, yet I am unable to find a mention of TPSR in the revision. Can the authors direct me to the line, Table or Figure in the paper in which "the comparison with TPSR" is made.

---

> ### Comment · Reviewer_SUeL · 2024-11-19
>
> > Bullet Point 1
>
> I have seen the revisions and while it brings about more clarity, it would be better if n is defined before or during the first mention of 'n-dimensional' in line 67.
>
> > Bullet Point 2
>
> I quote from the introduction of the paper:
>
> > Then, three key modules are proposed to enhance the approximation: (1) ...; (2) ...; (3) the Levenberg-Marquardt algorithm with L2 regularization, used for the adjustment of the expression structures and the optimization of constants.
>
> From reading this, it is fair to say that the paper claims the contribution of proposing the usage of the Levenberg-Marquardt algorithm to optimize constants for expressions in the SR algorithm. We know that this has been done from the paper by Kommenda that I cited in the review and this exact same issue and paper has been brought up by another reviewer as well. Thus, I mentioned that paper should be clearer on what is novel, which in my opinion has not yet been addressed even in the revision. To address this, the paper should rewrite their contribution so that it is not easily misinterpreted as claiming contribution by prior work. And also the paper should consider citing works, such as the paper by Kommenda, and discuss what is different.
>
> > Bullet Point 3
>
> I am confused by this reply because the paper cited is uDSR, but I only see uDSR results for SRSD and not SRBench. Is the paper missing a Figure?
>
> Additionally, if I follow the reasoning given in the reply, then why is Operon (which has closely overlapping R2 score, and also a non-overlapping model size) not considered when evaluating on SRSD. The difference in methods used for the SRBench and SRSD results is not consistent and disturbing.
>
> > Bullet Point 4
>
> This small Table provided is a good start, I would like to see the full results with error bars in the revision itself. TPSR is not included in the revision despite the reply and updated revision.
>
> > Bullet Point 5
>
> Likewise, good start, but I do not see the full results of ADGSGP and SLGP with error bars in the revision.
>
> > Bullet Point 6
>
> This reply is reasonable.
>
> > Bullet Point 7
>
> How about in comparison to the third-ranked method, Operon, which has a largely overlapping R2 score range with GESR and yet, has a much smaller, non-overlapping model size? Just a suggestion, there is a chance that a statistical test to compare methods may help, afterall, an error bar is useful and informative but not the best way to compare methods.
>
> > Bullet Point 8
>
> My concerns surrounding the topic of runtime has been sufficiently addressed by this reply.

---

> ### Author Response · Authors · 2024-11-21
> **Thank you for the continued discussion**
>
> > Reply to Comment 1:
>
> - The concern is that the line"..." is potentially misleading.
>
> Firstly, I don't think just the description divergence is enough to be a reason you set the contribution to be 'poor', when **there are two other main contributions and there is indeed a different role of the continuous optimization algorithm in our paper compared to that in the others**.
>
> **Secondly, the sentence you cited aims to describe the key components of our method and the corresponding main roles, rather than showing our contributions. The contributions are clearly claimed at the end of the Introduction section.** However, if the point you are stuck to is just I should not describe the constant optimization role of LM in the abstract, I will avoid mentioning the role of LM in optimizing the constants in the abstract, although it is also indeed helpful for constant optimization.
>
> In addition, there is a different role of the LM algorithm in our paper compared to that in the others and we have also clearly claimed many times in the reply and the main body of the paper that this one contribution is using it for the adjustment of the expression structures. Therefore, I don't think it will mislead readers.** If there is a paper that also claims to use the LM algorithm to optimize the expression structure, I will immediately revise the description.
>
>
> - 'The line "(3) We introduce the concept of using a continuous optimization algorithm to assist in adjusting the expression structure." is also potentially misleading in the same sense. '. To reiterate, the contributions listed do not accurately match what is done in the paper. :
>
> Since **we have emphasized and explained the role of adjusting the overall tree structure in lines 337~ 343 and lines 1000~ 1003 and the effect of using a continuous optimization algorithm to assist in adjusting the expression structure has already been shown in Appendix D.4** (the weight parameters can be optimized to zero to eliminate useless subtrees), you may have a misunderstanding of this paper and I recommend you read the main body of the paper first.
>
>
> > Reply to Comment 2:
>
> The example you have given in your previous comment that TPSR also supplements the E2E method in SRBench is inappropriate here because TPSR uses the E2E method as a backbone but there is nothing in common between our method and the TPSR, let alone that we have presented the comparison results with TPSR in the previous reply. As I have said before, whether a contribution is effective should depend on the novelty of a concept and the relative improvement compared to the same type of algorithms. Actually, the SBP-GP method is the most suitable baseline method to reflect the effectiveness of each contribution. The experiment results shown to you before are just the reply to your question. If you want me to present the experiment results in our manuscript, I will supplement it in Appendix D.7 of the later manuscript version. Hope that can satisfy you.

---

> ### Comment · Reviewer_SUeL · 2024-11-21
>
> > Firstly, I don't think just the description divergence is enough to be a reason you set the contribution to the 'poor', when there are two other main contributions and there is indeed a different role of the continuous optimization algorithm in our paper compared to the others.
>
> This is correct. However, the contribution score is not set to ‘poor’ because of the description divergence. There are multiple weaknesses of the paper which are mentioned in the review.
>
> > Secondly, what I want to do in the abstract is just to describe all the main roles of the key modules and we have clearly claimed the main contributions at the end of the Introduction section. I think you may know the difference between 'key modules' and 'contributions' right? Is there anything problem with presenting the key components of our method and describing all the corresponding roles? Especially there is indeed a different role of the LM algorithm in our paper compared to that in the others and we have also clearly claimed many times in the main paper that this contribution is using it for the adjustment of the expression structures, which is indeed one of the contributions. Or do you want to say that the LM algorithm is not one of our key modules or that the LM algorithm can not be used for the adjustment of the expression structures and constant optimization? In addition, while the main contributions have been clearly described at the end of the Introduction section (where I have specifically indicated the main role of continuous optimization algorithm and its specialty for our method), I don't think it will mislead readers. If you can find a paper that claims to use the LM algorithm to optimize the expression structure, I will immediately revise the description. Fine, if the point you are stuck to is just I should not describe the constant optimization role of LM in the abstract, I will avoid mentioning the role of LM in optimizing the constants in the abstract, despite it is also indeed helpful for constant optimization. Since we have emphasized and explained the role of adjusting the overall tree structure in lines 337~ 343 and lines 1000~ 1003 and the effect of using a continuous optimization algorithm to assist in adjusting the expression structure has already been shown in Appendix D.4 (the weight parameters can be optimized to zero to eliminate useless subtrees), I don't know why you think that doesn't match what we have done in the paper.
>
> I agree that ‘key modules’ alone may not imply ‘contributions’. However, the sentence ‘three key modules are proposed… (1)…, (2)…, (3)…’ reasonably implies ‘contributions’ because ‘key modules’ and ‘proposed’ are used in conjunction.
> The authors still do not seem to understand my concern. I will attempt to take a different approach by giving a suggested sentence instead.
>
> Suggested sentence: “The following are our key contributions: (1)…, (2)…, (3) We also introduce a new mechanism to reduce expression bloat by zero-ing the weights of uninformative subtrees (via LM algorithm).”
>
> Current sentences in the paper: "Then, three key modules are proposed to enhance the approximation: (1) ...; (2) ...; (3) the Levenberg-Marquardt algorithm with L2 regularization, used for the adjustment of the expression structures and the optimization of constants."
>
> "(3) We introduce the concept of using a continuous optimization algorithm to assist in adjusting the expression structure."
>
> I hope the authors can agree that the accuracy and clarity of their sentences can be improved. This is important because works like [1], which states in their abstract “…We parameterize symbolic metamodels using Meijer G-functions — a class of complex-valued contour integrals that depend on scalar parameters, and whose solutions reduce to familiar elementary, algebraic, analytic and closed-form functions for different parameter settings. This parameterization enables efficient optimization of metamodels via gradient descent, and allows discovering the functional forms learned by a machine learning model with minimal a priori assumptions…”, could also claim that they “introduce the concept of using a continuous optimization algorithm to assist in adjusting the expression structure”.
>
> Another example is [2], which can also claim the same.
>
> The point here is that by improving the listed contributions with higher clarity and better accuracy, it is more faithful to the contents of the paper and avoids making vague/broad contribution claims which overlaps with the claims of existing works.
>
> [1] Alaa, A. M., & van der Schaar, M. (2019). Demystifying black-box models with symbolic metamodels. Advances in neural information processing systems, 32.
>
> [2] Martius, G. S., & Lampert, C. (2017). Extrapolation and learning equations. In 5th International Conference on Learning Representations, ICLR 2017-Workshop Track Proceedings.

---

> > ### Comment · Reviewer_SUeL · 2024-11-21
> >
> > > The example you have given in your previous comment that TPSR also supplements the E2E method in SRBench is inappropriate here because TPSR uses the E2E method as a backbone but there is nothing in common between our method and the TPSR, let alone that we have presented the comparison results with TPSR in the previous reply. As I have said before, whether a contribution is effective should depend on the novelty of a concept and the relative improvement compared to the same type of algorithms. Actually, the SBP-GP method is the most suitable baseline method to reflect the effectiveness of each contribution. The experiment results shown to you before are just the reply to your question. If you want me to present the experiment results in our manuscript, I will supplement it in Appendix D.7 of the later manuscript version. Hope that can satisfy you.
> >
> > I strongly disagree that TPSR is inappropriate here. Following the same argument presented in the reply, all other methods in SRBench are irrelevant and GESR only need to compete with SBP-GP.
> >
> > GESR is an SR algorithm. It should be benchmarked against state-of-the-art SR algorithms, even those that use a different mechanism. TPSR and E2E are relevant benchmarks.
> >
> > Also, I would like to emphasize again that TPSR comparison is still not present at this time of writing. The authors evasive stance, redirecting me to a seemingly single-seeded run of TPSR in the reply while excluding any trace of TPSR in the main paper does not seem professional. If the comparison with TPSR is done, I would like to see it “in the paper”.

---

> > > ### Comment · Reviewer_SUeL · 2024-11-21
> > > **Comment on Edit**
> > >
> > > I realized the authors made an edit between the time I wrote a response and when their reply was first submitted. The content of the technical points they make are largely similar, so my response to their reply stays the same.

---

> > > > ### Author Response · Authors · 2024-11-22
> > > > **Thank you for the continued discussion**
> > > >
> > > > > Reply to Comment 1:
> > > >
> > > > - 'However, the contribution score is not set to 'poor' because of the description divergence. There are multiple weaknesses of the paper which are mentioned in the review.'
> > > >
> > > >
> > > > Then what are the other weaknesses of the contributions, especially when we have replied to most questions? The comments the reviewer has presented in the Weaknesses are almost something like the lack of experiments, the lack of comparison, and so on. I can not find any comment about the other two main contributions (not matter in the Strengths, the Weaknesses, or the Questions.  The only comment on the contributions is about the description divergence). The contributions proposed are even not mentioned in the reviewer's comments. It would be helpful if the reviewer can direct which comment is about the contributions. Otherwise, it seems the reviewer is attempting to weaken and deny all the main contributions of this paper by arguing over the clarity of the third contribution.
> > > >
> > > > > Reply to Comment 2:
> > > >
> > > > - "The authors still do not seem to understand my concern. I will attempt to take a different approach by giving a suggested sentence instead. Suggested sentence: “The following are our key contributions: (1)…, (2)…, (3) We also introduce a new mechanism to reduce expression bloat by zero-ing the weights of uninformative subtrees (via LM algorithm).”"
> > > >
> > > > I do agree that it is important to be accuracy and clarity for a paper. However, the core of this contribution is the capability to adjust the tree structure to maximize the probability of not missing the solution when the symbolic tree includes all the neccessary elements. Reducing expression bloat is merely a secondary effect, and your previous description was not sufficiently accurate to conclude this contribution (also thanks for your detailed suggestion). The description "the Levenberg-Marquardt algorithm with L2 regularization, used for the adjustment of expression structures and the optimization of constants" does accurately highlight both the unique role of the LM algorithm in this paper and its indispensable function in our semantic approach (adjusting the tree structure and assisting in optimizing global constants, especially considering the large number of constants compared to traditional methods). Nevertheless, considering that the LM algorithm does need to combine the context of the proposed semantic method to reflect its particularity and the controversies surrounding constant optimization, I have revised it to "the Levenberg-Marquardt algorithm with L2 regularization, used for the adjustment of expression structures and the balance of global subtree weights to assist the proposed geometric semantic search operator." in the updated manuscript.
> > > >
> > > > - ...(3) We introduce the concept of using a continuous optimization algorithm to assist in adjusting the expression structure." I hope the authors can agree that the accuracy and clarity of their sentences can be improved. This is important because works like [1], which states in their abstract “…We parameterize symbolic metamodels using Meijer G-functions — a class of complex-valued contour integrals that depend on scalar parameters, and whose solutions reduce to familiar elementary, algebraic, analytic and closed-form functions for different parameter settings. This parameterization enables efficient optimization of metamodels via gradient descent, and allows discovering the functional forms learned by a machine learning model with minimal a priori assumptions…”, could also claim that they “introduce the concept of using a continuous optimization algorithm to assist in adjusting the expression structure”.
> > > >
> > > >
> > > >
> > > > That's right. There is indeed inaccuracy in the newly added description "We introduce the concept of using a continuous optimization algorithm to assist in adjusting the expression structure". Referencing the reviewer's suggestion, I have revised it in the updated manuscript, which is "We also present a mechanism to assist the geometric semantic method in adjusting the generated expression structure and capturing the potential solution by zero-ing the weights of uninformative subtrees (via the LM algorithm).". Thanks for your suggestion.

---

> > > > > ### Author Response · Authors · 2024-11-22
> > > > > **Thank you for the continued discussion**
> > > > >
> > > > > > Reply to Comment 3:
> > > > >
> > > > > What I want to emphasize is that the current results are enough to demonstrate the effectiveness of the three contributions. The sota is important, which can demonstrate the potential and the power of the method in this symbolic regression field. However, whether it is compared to the methods the reviewer points out or not should not be the strong reason the reviewer disagrees with the effectiveness of the three contributions.  It is also important to make a multifactorial evaluation of the novelty of the methods and concepts proposed and reference the relative improvement compared to the same type of algorithms. Besides, from the argument the reviewer presents in the comment, should TPSR also compare to the uDSR first instead of E2E, otherwise the reviewer will also not recognize the effectiveness of their contributions? By the way, there is no need for the 'evasive stance' since the original intention of presenting the additional results is just to reply to your question. The experimental results presented in the reply are based on the mean results with three random seeds, which is enough for only discussion. If we include the TPSR in Figure 5 of the paper, it would take lots of time to supplement the additional noise experiments and increase the number of seeds. However, considering your partial comments and the possibility of your future update to score, There is just a lack of motivation to rush into the task.

---

### Official Review · Reviewer_gDcq · 2024-11-04

**Soundness:** 2
**Presentation:** 1
**Contribution:** 3
**Rating:** 5
**Confidence:** 3

**Summary:**

This paper identifies a problem with the prevailing genetic programming (GP) methods for symbolic regression: the random cross-over policy adopted makes GP less effective in generating promising formulas. Consequently, the paper introduces a semantic geometric evolution algorithm, which maps the expression search space into a target semantic search space to approximate the mutation operations. The semantic gradients and the geometric semantic method are proposed to help guide search in the high dimensional sub-semantic search space for mutations, with the Levenberg-Marquardt algorithm with L2regularization adopted for constants optimization and local structure adjustments.

**Strengths:**

**1)** The motivation for the paper is clear, and the semantic geometric evolution algorithm is an interesting idea to incorporate guidance for mutations.

**2)** The proposed semantic gradients address the inconsistency of the sub-semantic space and the target semantic space.

**3)** Conducted experiments on well-recognized SRBench and other datasets. Ablation studies support the claimed ideas.

**Weaknesses:**

**1)** The definitions for the notations and terminologies are ambiguous and missing, making the methodology part hard to follow. For example:
 ***1.*** In line 159 "n-dimensional semantic space", what does $n$ represent here? Is it the dimension of the feature or the dimension of the train set as explained in line 249? Same question as in equation (2).
***2.*** In line 187, what does  $s_i$ represent here? I understand it may refer to the semantics of the subtree, but only list subtree $tr$ may be misleading.
***3.*** In line 232, what is the dimension of the subtree? Is it the same dimension of $n$?
***4*** Notation conflicts for $k$ in 269 "top-k" candidates and equation (7)
***5.*** In equation (11), is the summation over $j$? In line 324 and equation (11), what do $i$ and $j$ represent here as the superscript?

**2)**  A typo in line 269: following m subtree(subtrees) for each $tr_1$ is(are) chosen to form the candidate set.

**3)** The benchmark baseline models are not state-of-the -art.

**Questions:**

**1)** Can you provide explanations concerning the above notations and terminologies?

**2)** How do you obtain the sub-target semantics $st$ for the loss calculation in line 274 as well as the semantic gradients in equation (3)?

---

> ### Author Response · Authors · 2024-11-19
> **Rebuttal by Authors**
>
> Clarification and explanation for weakness:
>
> 1.	Thanks for the suggestions about the presentation.
> (1) For the meaning of ‘n’: yes, like we have claimed in line 249, The value ‘n’ is equal to the training set size, that is why we say semantic space is a sparse space and is also the reason we present the geometric semantic method to make quick approximation in semantic space. For a better understanding, we now add the corresponding description in line 159;
> (2) Thanks for your suggestion, now we have added $s_i$ after ‘the semantics’ in line 187 to make it more explicit, and the relationship between $s$ and $tr$ is equivalent to the relationship between the semantics and its corresponding symbolic representation. By the way, the notations in our manuscript are actually unified, and we have previously described this notation in lines 162~163 of the original manuscript.
> (3) Yes, that is because the dimension of the chain rule should be equivalent to the dimension of the semantic space.
> (4) Actually, I think top-k can be considered a commonly used term, but you are right. For the sake of rigor, we have revised it to 'top-t' in the updated manuscript.
> (5) Thanks for your suggestion, $M = \max_{0\leq i\leq n} r_i = \max_{0\leq i\leq n} ((\mathbf {st}^i - \mathbf {s}^i)^2\cdot\left|\nabla T\right|_{\mathcal{N}}^i)$, and we have revised this error in the updated manuscript;
>
> 2. Thanks for your careful check on the typo errors, we have gone through the entire manuscript carefully and made the corresponding corrections in the updated manuscript.
>
> 3. Despite the emergence of some new methods recently, most of them still use the SRBench benchmark as the main baseline. It can be easily found out that our method is still competitive compared to the accuracy performance they have reported on the SRBench benchmark. Furthermore, our method still exhibits competitive performance on the SRSD benchmark along with some recently proposed methods such as uDSR.
>
> Answer for Questions:
>
> 4. Please see reply 1.
>
> 5. We have updated some figures and descriptions in the updated manuscript for better understanding. The sub-target semantics are computed through semantic backpropagation, which has been described in lines 177~180 of the original manuscript. The backpropagation process is also briefly shown in the 'Semantic Backpropagation' of Figure 1.

---

> > ### Comment · Reviewer_gDcq · 2024-11-25
> >
> > 1. Thank you for addressing the concerns regarding notations and modifying the manuscript. While these changes improved my understanding of the methodology section, the organization of the notations could still be clearer. For example, in line 259, it would be more effective to position the syntax closer to its definition, such as: "migrate the distribution of candidate semantics $s'$ to the sub-target semantics $st$." Without this adjustment, readers might struggle to identify which term represents the candidate semantics in Equations (4) and (5).
> >
> > 2. The syntax currently employs a complex system of subscripts and superscripts, and several symbols appear without explicit definition (e.g., notations in Figure 1). To improve clarity and accessibility for readers, I strongly recommend including a notation table or glossary that comprehensively defines all variables and symbols used throughout the paper. This addition would significantly enhance the paper's readability.
> >
> > 3. Thank you for clarifying the questions related to the benchmarks. Beyond the consistency issues raised by reviewer ZPa5 regarding SR baselines, I have a few additional observations:
> >   - From Figure 4 (model size) and Table 9 (appendix), it seems that GESR tends to generate formulas that are notably large and lengthy, even after simplification. This might suggest a tendency for GESR to produce high-complexity expressions that may indicate overfitting to the Feynman problems.
> >   - While this behavior might be acceptable if accuracy is the primary evaluation metric, it becomes problematic when using recovery rate, which requires the generated expressions to match the exact ground truth form.
> > To ensure fair comparisons, I suggest evaluating GESR and the other baselines with recovery rate as an additional metric alongside accuracy. This would provide a more balanced assessment of model performance.
> >
> > Based on the current presentation quality and evaluation design, I will maintain my score for now. However, addressing the concerns raised about notations and benchmarks could strengthen the paper further.

---

> > > ### Author Response · Authors · 2024-11-25
> > > **Thanks for the the continued discussion**
> > >
> > > Thanks for your comment. Your suggestions are indeed helpful for improving the presentation. The relevant revision has been done in the updated manuscript and the notation table is also presented in Appendix C.3 of the manuscript.
> > >
> > > As for the benchmark, the symbolic solution rate (recovery rate) has already been presented in Appendices D.3 and D.6 of the original manuscript, and Table 10 (formerly Table 9) also includes some examples of the symbolic solutions, although the main purpose of this method is to achieve high accuracy under the constraint of a certain model complexity instead of the recovery rate. The relevant analysis has also been presented in lines 1389~1428 and lines 966~971.

---

> ### Comment · Reviewer_gDcq · 2024-12-02
>
> Thank you for your clarification! I would recommend citing the notation table at the beginning of the methodology section to help readers locate it. From Table 8 and Table 9, I observed that the proposed GESR suffered from complexity issues for recovery rate evaluation on easy tasks. Overly complicated solution expressions would make it hard for researchers to discover vital underlying dependency relationships in real-world applications, and I think the author can further improve their work with better complexity control for the robust performances across different tasks and real-world significance.

---

> ### Author Response · Authors · 2024-12-03
> **Thanks for your continued discussion**
>
> Thanks for your suggestion. However, since the deadline for manuscript revisions has passed, we will not make changes at this time. By the way, from the notations we have already listed in the manuscript, it is easy to observe that both the symbols and the subscript and superscript system (subscript for dimension indices and superscript for identifying each entry) maintain consistency throughout the manuscript. Additionally, relevant descriptions have been provided before the first appearance of each symbol. Therefore, placing the notations list in the appendix (or wherever appropriate) is also sufficient.
>
>
> > For the comment about Table 8 and Table 9:
>
> Thanks for your comment. Interpretability is indeed an important topic. However, it should be clarified that the recovery rate is not equivalent to the complexity. As observed in Table 10, even when symbolic solutions are not found, the simplified accuracy solutions we have listed, mostly fall within the range of approximately 10 to 40 size (except for the unfitted instances, some specific conditions have been discussed in lines 1329-1338). In fact, such size is already competitive and interpretable, especially compared with semantic methods.
>
> Then, while the primary focus of this paper is on accuracy, it is worth noting that the recovery rate is also competitive. The symbolic solution rate on medium and hard datasets, as presented in Tables 8 and 9, is the highest. Our method also ranks second on easy datasets without dummy variables. Only the recovery rate performance on easy datasets with dummy variables is relatively less impressive, which may indeed show the relatively limited resistance to dummy variables since we use L2 regularization to resist potential noise instead of L1.
>
> Thanks for the suggestions you have given.

---

### Note · Authors · 2024-12-25

I have read and agree with the venue's withdrawal policy on behalf of myself and my co-authors.